https://doi.org/10.1038/s41467-021-25086-5　　**OPEN**

# Inferring multilayer interactome networks shaping phenotypic plasticity and evolution

Dengcheng Yang[1,2], Yi Jin[1,2], Xiaoqing He[1,2], Ang Dong[1,2], Jing Wang[1,2] & Rongling Wu [1,2,3 ✉]

Phenotypic plasticity represents a capacity by which the organism changes its phenotypes in response to environmental stimuli. Despite its pivotal role in adaptive evolution, how phenotypic plasticity is genetically controlled remains elusive. Here, we develop a unified framework for coalescing all single nucleotide polymorphisms (SNPs) from a genome-wide association study (GWAS) into a quantitative graph. This framework integrates functional genetic mapping, evolutionary game theory, and predator-prey theory to decompose the net genetic effect of each SNP into its independent and dependent components. The independent effect arises from the intrinsic capacity of a SNP, only expressed when it is in isolation, whereas the dependent effect results from the extrinsic influence of other SNPs. The dependent effect is conceptually beyond the traditional definition of epistasis by not only characterizing the strength of epistasis but also capturing the bi-causality of epistasis and the sign of the causality. We implement functional clustering and variable selection to infer multilayer, sparse, and multiplex interactome networks from any dimension of genetic data. We design and conduct two GWAS experiments using *Staphylococcus aureus*, aimed to test the genetic mechanisms underlying the phenotypic plasticity of this species to vancomycin exposure and *Escherichia coli* coexistence. We reconstruct the two most comprehensive genetic networks for abiotic and biotic phenotypic plasticity. Pathway analysis shows that SNP-SNP epistasis for phenotypic plasticity can be annotated to protein-protein interactions through coding genes. Our model can unveil the regulatory mechanisms of significant loci and excavate missing heritability from some insignificant loci. Our multilayer genetic networks provide a systems tool for dissecting environment-induced evolution.

[1] Beijing Advanced Innovation Center for Tree Breeding by Molecular Design, Beijing Forestry University, Beijing, China. [2] Center for Computational Biology, College of Biological Sciences and Technology, Beijing Forestry University, Beijing, China. [3] Center for Statistical Genetics, Departments of Public Health Sciences and Statistics, The Pennsylvania State University, Hershey, PA, USA. ✉email: rwu@bjfu.edu.cn

Most organisms are equipped with a capacity to produce multiple phenotypes in response to environmental change[1–6]. This so-called phenotypic plasticity is coded by a plastic developmental program that allows the organisms to sense environmental cues in early stages of life and develop phenotypes better adapted to environments encountered later in life[7]. Phenotypic plasticity is thought to be under genetic control[8–12]; i.e., specific genes may occur to govern its pattern, sign, and magnitude. In modern ecological, evolutionary, and medical genetics, the genetic mechanisms underlying phenotypic plasticity and its evolutionary novelty have emerged as an important topic of research in both theory and application[4,13–18]. Genetic mapping and genome-wide association studies (GWASs) have been used to map the genome-wide distribution of quantitative trait loci (QTLs) responsible for phenotypic plasticity in a variety of species[19–25]. With the advent of high-throughput genotyping techniques, there is a pressing demand to chart a more detailed and precise genetic atlas of this phenomenon.

Existing approaches are founded on reductionist thinking, which can only identify individual key QTLs at a time. However, it is becoming increasingly clear that a deeper genetic understanding of phenotypic plasticity as a complex trait requires not only a detailed characterization of its underlying individual genes, but also of their interactions as a cohesive whole[26–31]. A recently emerging theory, known as omnigenic theory, states that complex traits are controlled by all genes carried by an organism[32]. Taken together, to better interpret phenotypic plasticity, we need to coalesce all genes into genetic interaction networks and trace the roadmap of each gene toward this phenomenon. Networks are mathematical graphs, connecting nodes (e.g., genes) via edges (genetic interactions), which are particularly powerful for disentangling complex systems[33,34]. However, there is a formidable challenge in reconstructing genetic networks. First, for ordinary GWAS, hundreds of thousands of SNPs genotyped for each sample are not uncommon, making it computationally infeasible to reconstruct an omnigenic network. Second, the detection of genetic interactions requires a sample size that is hardly met in practice. Computer simulation shows that as many as 10,000 samples are needed to detect genetic interactions between a pair of genes in a GWAS[35], let alone millions of millions of gene pairs.

Third, in classic quantitative genetics, genetic interactions are defined as epistasis, which describes how much the effect of one gene reciprocally depends on the other gene[34,36–38]. Statistical approaches currently used in the literature cannot reveal the casualty of epistasis and the sign of casualty. Understanding the direction of regulation from one gene to next is of utmost significance to disentangle the genetic architecture of complex traits and design an efficient and effective gene editing program for trait improvement and disease control. Fourth, genetic variation in phenotypic plasticity is displayed as gene–environment interactions of which most studies are based on linear additive models[39]. However, the organism often responds to environmental change in a nonlinear manner owing to substantial uncertainties and random effects across timescales[40–42]. Nonlinearities between genes and environment can help trait phenotypes maintain their robustness to random perturbations in environmental exposures[43].

Here, we develop a computational model to infer omnigenic interactome networks underlying phenotypic plasticity. The model absorbs and integrates the elements of multiple previously disjointed disciplines into a unified framework. Association analysis based on individual SNPs estimates the marginal (net) genetic effect of each SNP on a complex trait from a pool of genome-wide markers. Evolutionary game theory[44] allows us to interpret how the genetic effect of a SNP is determined by its own intrinsic capacity and the epistatic influences of other loci on it.

The integration of evolutionary game theory with predator-prey theory[45–49] makes it possible to derive a generalized nonlinear Lotka–Volterra (nLV) equations that decompose the net genetic effect of a SNP into its two components, the independent effect due to its intrinsic capacity and the dependent effect resulting from regulation by other SNPs[50]. According to network theory, we encapsulate independent effects of individual SNPs as nodes and dependent effects of SNP pairs as edges into a graph, in which edges represent bidirectional, signed, and weighted (bDSW) genetic interactions (i.e., epistasis). Viewing all SNPs together as a quantitative system, the distribution of bDSW interactions portrays a picture of the biological role of epistasis. By incorporating developmental modularity theory[51] and high-dimensional statistical models, we can reconstruct multilayer, multiplex, large-scale, and sparse genetic networks from any number of SNPs.

We design and conduct two independent GWAS experiments of phenotypic plasticity to abiotic and biotic factors, respectively, using bacterial species *S. aureus*. As one of the most important members of the Firmicutes, *S. aureus* often cause skin infections, but can also lead to pneumonia, heart valve infections, and bone infections[52]. The abiotic GWAS experiment is to study how the growth of *S. aureus* responds to vancomycin, a glycopeptide antibiotic designed to control and treat *S. aureus*-caused diseases by inhibiting cell wall biosynthesis[53,54]. In clinical practice, the use of this drug has incurred vancomycin-intermediate *S. aureus* (VISA) isolates, with the minimal inhibitory concentration (MIC) of 4–8 μg/mL[55]. Vancomycin non-susceptibility in *S. aureus* involves a genetic component. Several nucleotide substitutions that distinguish the vancomycin-susceptible *S. aureus* from VISA isolates have been identified by a whole-genome sequencing approach[56–58]. Some studies further used GWAS to characterize common genetic variants associated with antibiotic resistance in *S. aureus*[55,59]. However, despite these advances, a comprehensive portrait of the genetic control mechanisms underlying vancomycin resistance is still unclear. The biotic GWAS experiment aims to investigate how *S. aureus* changes its growth when encountered with another species, *Escherichia coli*. Studying the plastic response of one bacterial species to its co-existing species has become increasingly interesting to researchers from many fields including the gut microbiomes[60,61], but the genetic architecture of this phenomenon has been little explored. Jiang et al[60]. characterized a set of specific QTLs in *S. aureus* that mediates the change of microbial growth from a socially isolated environment to a socialized environment. It is likely that phenotypic plasticity to biotic factors includes a complex, still unknown genetic machinery.

We apply our new model to systemically dissect the genetic architecture of how *S. aureus* responds to an abiotic factor (drug) and biotic factor (species coexistence). Phenotypic plasticity can be quantitatively defined as the difference of phenotypic values for the same genotype expressed in two different environments[9–11]. We define the difference of microbial growth for the same *S. aureus* strain expressed in vancomycin-free and vancomycin-exposed media or in isolated and *E. coli*-socialized media as the abiotic or biotic phenotypic plasticity of this strain, respectively. First, we globally view the genetic landscape of each type of phenotypic plasticity, filled by a complete set of bDSW interactions, from hundreds of thousands of SNPs and delimit this landscape into functionally distinct network communities. Second, we characterize the detailed roadmap through which individual key genes determine phenotypic plasticity directly and/or through multiple indirect pathways. Our model opens up a new avenue to unveil the genetic complexities of how genes interact with the environment to help the organism better adapt to environmental and biological cues.

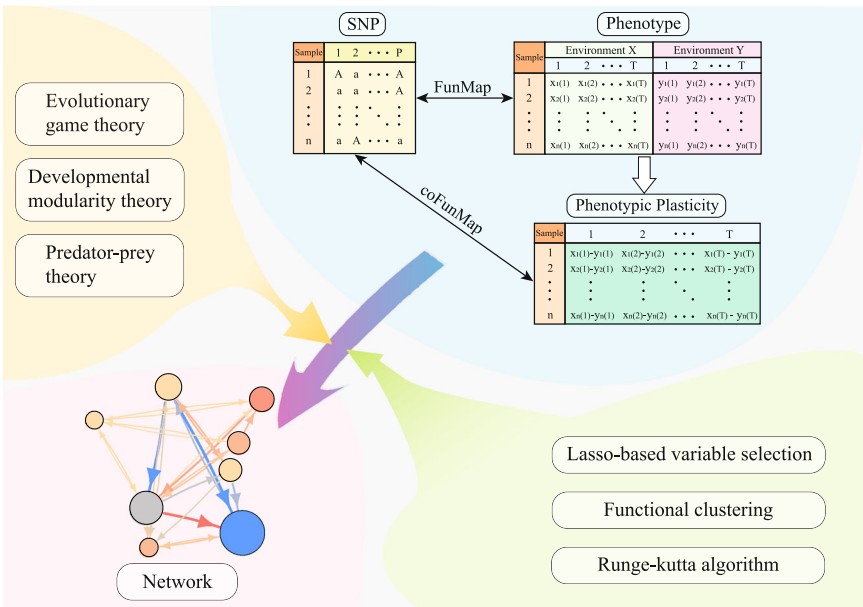

**Fig. 1 Flowchart of model derivation with genetic data and time-series phenotypic data in two environments X and Y as inputs.** Environment-specific phenotypes are used to calculate phenotypic plasticity, which is mapped by coFunMap, an extension of single-trait functional mapping (FunMap) to a composite trait. coFunMap is integrated with elements of evolutionary, ecological, and developmental disciplines to reconstruct maximally informative (bidirectional, signed, and weighted) interaction networks by implementing high-dimensional statistical and mathematical methods. In the network, line arrow represents the direction of regulation by one SNP (node) to the next, line color denotes the sign of regulation (warm for promotion, cold for inhibition), and color metric indicates the strength of regulation.

## Results

We derive a computational model for reconstructing omnigenic interactome networks underlying phenotypic plasticity. We outline the procedure of model derivation in Fig. 1, starting with genotype data and phenotype data collected in two environments from a general GWAS. The new model is the seamless integration of traditional mapping approaches and multiple elements of different disciplines through high-dimensional statistical and mathematical reasonings. A detailed description of model development is given in the "Methods".

**A standard procedure of QTL mapping**. To show how our model works in the practical dissection of phenotypic plasticity, we focus on the presentation of results from the abiotic GWAS experiment, followed by the biotic GWAS experiment. In the abiotic experiment, 99 *S. aureus* strains were cultured in vancomycin-free medium (control) and vancomycin-exposed medium (stress). The mean abundance growth of these strains obeys an S-shaped logistic equation, as observed in many previous studies[62], but the amount and pattern of growth vary from control to stress (Fig. 1a). In general, strains grow much faster ($r = 0.24$ vs. $0.16$), more rapidly reach maximum relative growth rate ($t_I = 12.1$ h vs. $21.4$ h), and displays greater asymptotic growth ($a = 1.54$ vs. $1.10$) in control than stress conditions. Increasing variability in growth trajectory is produced when strains are exposed to vancomycin. In contrast to all individual strains displaying S-shaped growth curves in control, a portion of strains in stress grow a little, having high susceptibility to vancomycin. Those that grow in an S shape in stress imply their high vancomycin resistance, marginally associated with an elevated parental MIC ($P = 0.12$) (Table S1) and particular evolutionary groups (Fig. S1A). By plotting microbial abundance in control against that in stress across time points (right panel, Fig. 2a), we illustrate an overall picture of how the same strain changes its growth trajectory in response to vancomycin, i.e., developmental

phenotypic plasticity, and how the capacity of this plasticity is strain-dependent.

Unlike a general complex trait, developmental phenotypic plasticity is a composite trait derived from phenotypic values at multiple time points expressed in different environments. To better dissect it, we implement composite functional mapping (coFunMap), an extended model of functional mapping. By analyzing microbial growth trajectories that have been adjusted for population structure (Fig. S2), coFunMap identifies 16 significant SNPs (i.e., QTLs) that control plastic response to vancomycin (Figs. 2b, 1c) from a total of 25,173 SNPs. Of these QTLs detected, 11 are annotated to candidate genes with specific cellular functions and the remaining 5 are located in non-coding regions (Table S2). Two annotated QTLs Q550312 and 550323 reside in the region of gene *SAOUHSC_00544* (*SdrC*) that codes fibrinogen-binding protein. In a genomic analysis of highly vancomycin-resistant *S. aureus* strains developed by in vitro mutagenesis, gene *SdrC* was identified to play an important role in vancomycin resistance[63,64]. QTL Q25261 is located at gene *SAOUHSC_00020* (*walR* or *yycG*) encoding cell envelope biogenesis[65]. *walRK* phosphorylation activates the expression of *YycH* and *YycI* to control cell wall metabolism and biofilm formation in *S. aureus*[66,67].

We further estimate the genetic effect curves of each QTL and tested in which temporal pattern a QTL affects the phenotypic plasticity of microbial growth (Fig. 2d). All QTLs start to act from the time of culture and then change their genetic effects in a cyclic manner. They first quickly increase their effects to reach a high peak at time 10 h after culture, followed by a dramatic decrease, after which a new cycle, but to a lesser extent, is started. As a comparison, we randomly choose five insignificant SNPs from the bottom of the Manhattan plot (Fig. 2b) and find that their effect curves are close to zero line, implying their negligible role in shaping phenotypic plasticity.

The analytical procedure described above represents a standard flowchart of QTL detection from genetic mapping or association

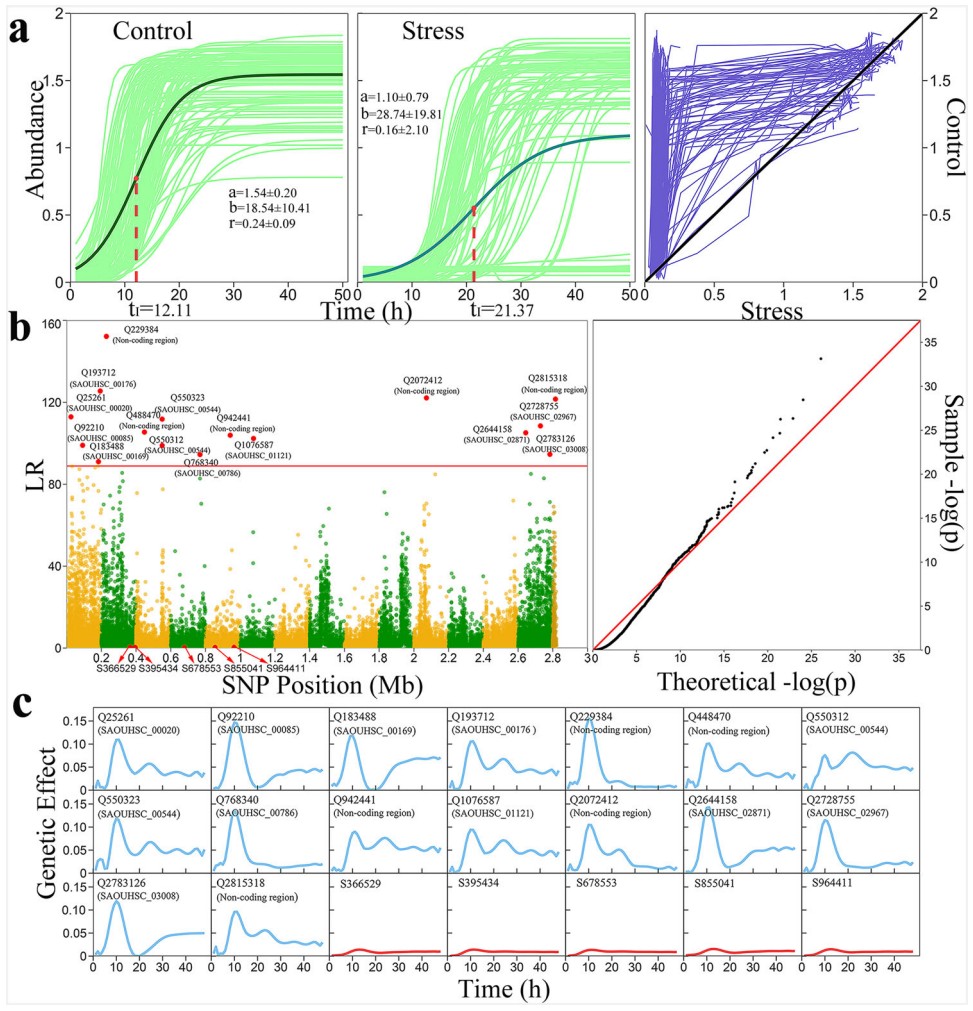

**Fig. 2 A general flowchart of genetic mapping for dynamic complex traits. a** Growth trajectories of 99 *S. aureus* strains (gray think lines) cultured in vancomycin-free (control, 0 μg/mL) and vancomycin-stress (6 μg/mL) media. The mean growth is fitted by a logistic equation (think green line), from which the timing of maximum growth rate $t_l$ is estimated. The square plot of strain growth in control against stress is shown, where the degree of deviation from the diagonal line is positively associated with vancomycin-induced difference in growth curve. **b** Left panel: Manhattan plot of the significance test based on log-likelihood ratio (LR) for associations between the phenotypic plasticity of microbial growth trajectory and SNPs distributed on the *S. aureus* genome. coFunMap identifies 16 significant SNPs (called QTLs) that are indicated above red horizontal line representing the genome-wide critical threshold determined from 1000 permutation tests. At the bottom of the Manhattan plot, five insignificant SNPs that are chosen for subsequent dissection are indicated. Right panel: Q–Q plot used to characterize the extent of deviation of the observed distribution of the test statistic from the expected (null) distribution for phenotypic plasticity-related GWAS of 99 *S. aureus* strains. **c** Curves of genetic effects on phenotypic plasticity for 16 QTLs and five chosen insignificant SNPs estimated by coFunMap.

studies. Our model can address several unanswered questions: (1) how and why significant QTLs are significant, (2) why insignificant SNPs are not significant, and (3) whether insignificant SNPs can become significant. In the end, we will depict the functional roadmap of each SNP towards phenotypic plasticity.

**Detection of network communities**. To systematically characterize the role of each SNP in mediating phenotypic plasticity, an appropriate approach is to reconstruct a genetic network that covers all pairwise links of SNPs considered. However, for a GWAS with thousands of thousands of SNPs, this is computationally prohibited. More importantly, full interconnections of genes do not exist in a cell since this violates the basic biological rule, i.e., buffering against random perturbations through sparsity[68–71]. Developmental modularity theory states that a complex system is often broken down into multiple distinct network communities within which components are connected more tightly with each other than those from other communities[72,73]. We introduce this theory to divide a large-

scale network into its delimited communities in which genes are linked more tightly with each other than with those from other communities[74]. It is possible that genes with the same community follow a similar temporal pattern of genetic effects on phenotypic plasticity, as previously detected in a longitudinal GWAS[75,76]. We use coFunMap to estimate genetic effect curves for each SNP and then implement functional clustering (FunClu)[77,78] to classify all SNPs into distinct modules (i.e., communities) based on the similarity of their effect curves.

We identify 14 modules (labeled as M1–M14) of different sizes among 25,173 SNPs based on Bayesian information criterion (BIC), with each module displaying a different pattern of genetic effects (Fig. 3a). The overall genetic effects of most modules cyclically change with time, but some modules, like M4 and M9, display small constant genetic effects. Modules M1, M2, M5, and M11 tend to increase their genetic effects with time, despite to different extents, leading to larger effects detected at the late stage of culture than at the early stage. Taken together, we detect 14

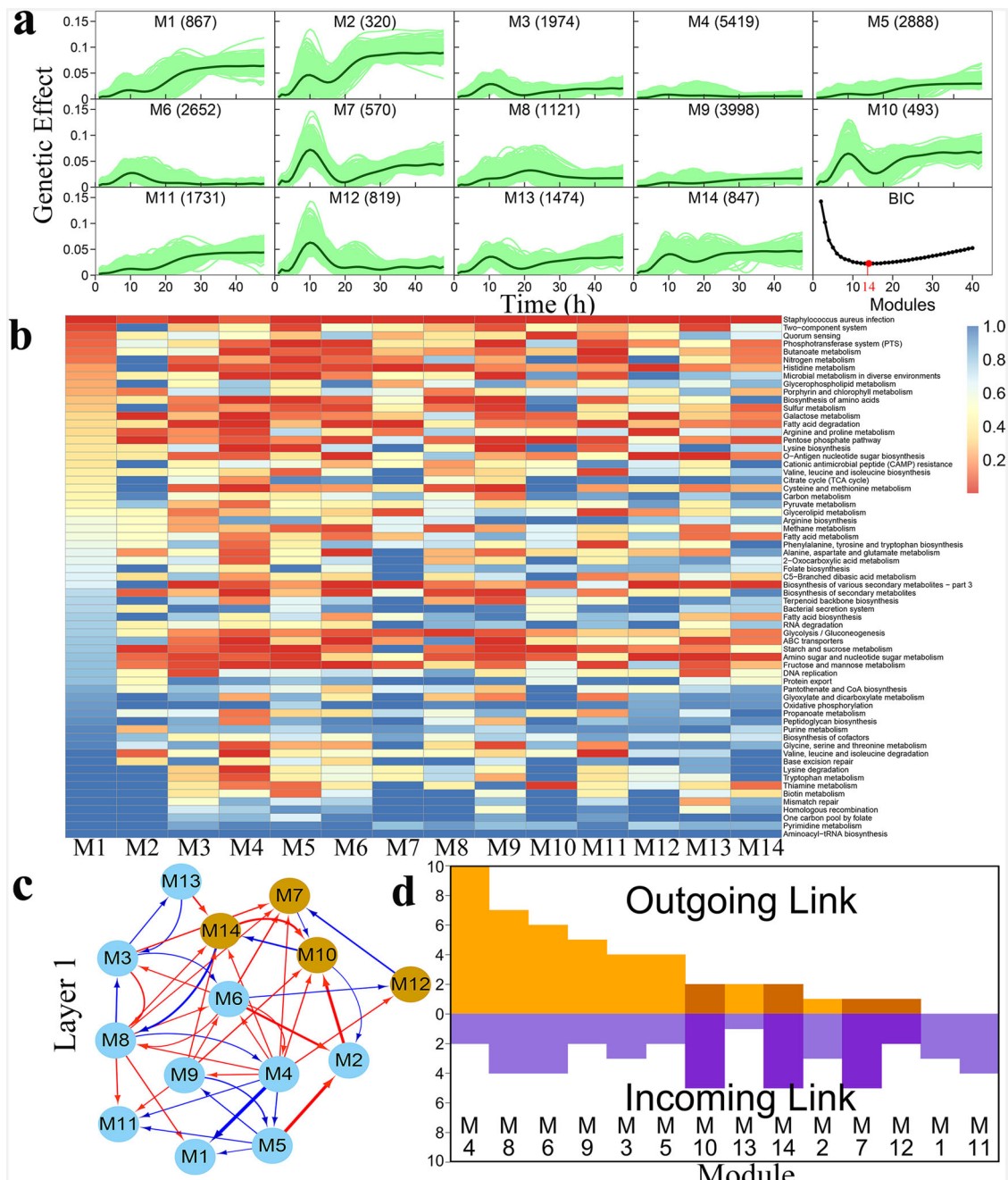

**Fig. 3 Identification of network communities from a large-scale network. a** The curves of genetic effects on the phenotypic plasticity of microbial growth trajectory for 14 distinct modules classified from 25,173 SNPs according to BIC. Thick green line represents the mean curve of all SNPs within modules, with individual SNP-specific curves denoted by thin green lines. **b** Heatmap of gene functions from 14 modules by gene set enrichment analysis. **c** The 14-node inter-module network displayed at the first layer of the whole network. QTLs detected by coFunMap are sorted into modules M7, M10, M12, and M14, highlighted in orange. Red and blue arrowed lines stand for the direction of up-regulation and down-regulation, respectively, with the thickness of lines proportional to the strength of regulation. **d** Distribution of the number of outgoing and incoming links over different modules. The positions of QTL-containing modules are indicated.

distinct network communities that form a 14-node genetic network for the plastic response of *S. aureus* to vancomycin. The 16 QTLs detected by coFunMap are distributed in communities M7, M10, M12, and M14, rather than randomly distributed in different communities, suggesting that QTLs detected tend to function in a similar manner. This argument is strengthened by the fact that the QTL-containing communities, especially M7, M10, and M12, display a similar pattern of time-varying genetic effects. However, gene enrichment analysis

annotates these QTL modules with different biological functions (Fig. 3b). For example, M7 contains genes coding histidine metabolism, fatty acid degradation, glycerolipid metabolism, and biosynthesis of various secondary metabolites, M10 involves genes associated with pentose phosphate pathway, quorum sensing, thiamine metabolism, starch, and sucrose metabolism, and M12 contains genes with functions in galactose metabolism, arginine and proline metabolism, O-antigen nucleotide sugar biosynthesis, biosynthesis of various secondary metabolites, and

amino sugar and nucleotide sugar metabolism. Taken together, although individual genes function differently, their combination may generate a similar influence on the phenotypic plasticity of *S. aureus* to vancomycin stress.

**Multilayer genetic networks**. We divide a large-scale network into 14 communities, further divide each community into distinct subcommunities and divide each subcommunity into distinct sub-subcommunities, and repeat this process until no communities at the smaller level can be detected. The interconnections of different communities from top to bottom levels form a multilayer, multiplex network that records the pathway of each gene toward phenotypic plasticity. We take the means of genetic effects over all SNPs within modules and treat these means as different game players of a system. We implement evolutionary game theory[44] to model how one module mean is affected by its own strategy and the strategies of other module means. According to this argument, we reconstruct a 14-node inter-module network, which represents a top genetic network at the first layer (Fig. 3c). Of all 182 possible pairwise regulations, we only identify 45 links (24.4%) within the macroscopic network, showing the sparsity of the network. There are a few pairs of SNPs that reciprocally regulate each other, but most links are directional, consistent with the rock-paper-scissor rule for animal asymmetrical interactions detected in macro-communities[79]. A node is said to be outgoing if it regulates others or incoming if it receives regulation by others. The number of outgoing links gradually decreases from 10 (M4) to 0 (M1 and M11), but incoming links are more uniformly distributed among modules (Fig. 3c). We call those modules that have a large number of outgoing links "leaders." From gene enrichment analysis, it is interesting to see that the leader M4 performs a much wider spectrum of biological functions, compared to subordinates M1 and M11 that only receive incoming links (Fig. 3b). Four QTL-containing modules M7, M10, M12, and M14 tend to receive incoming links (including those from the leader) rather than exert outgoing links, each of which performs a different set of biological functions. These findings suggest that QTLs detected by existing mapping approaches may be located in downstream, whose function relies on the regulation of leaders.

Under the first-layer network at the module level (Fig. 3c), we reconstruct second-layer inter-submodule networks for each module. We classify SNPs in QTL-containing modules M4, M7, M10, and M12 into submodules that are used to reconstruct the second-layer networks (Figs. 4, S3–S5). In the second-layer 35-node network of M12, promotion is more common than inhibition, and outgoing links are more than incoming links (Fig. 4a). QTL-containing submodules SM25, SM17, and SM29 tend to receive incoming links than exert outgoing links. We reconstruct SNP-SNP networks at the third layer for QTL-containing submodules SM17, SM29, and SM25 (Fig. 4b), from which we characterize how each QTL triggers its effect on phenotypic plasticity. We perform pathways analysis for three SNP networks using software STRING[80] to reconstruct downstream protein-protein networks (Fig. 4c).

In the SM29 SNP network, S95811 (residing in gene *SAOUHSC_00090*) is a hub with more links than average, and its coding protein AID38557.1 also serves as a hub in the protein network. In the same network, S95811 interacts with S94818 (residing in *SAOUHSC_00089*) and, this interaction code the interaction between two proteins AID38557.1 and AID38556.1, each coded by one of these two SNPs. In the SM25 SNP network, two linked SNPs S319914 and S324256 code the interaction between their coding proteins AID38813.1and AID38818.1 (Fig. 4c). S2728755 (located at *SAOUHSC_02967*) serves as a hub in the SM17 SNP network and, correspondingly, protein

AID41345.1 it codes is a hub in the protein network. Also, the interaction of S2728755 with S2748895 (located at *SAOUHSC_02982*) coding protein AID41358.1 is reflected by the link of their coding proteins (Fig. 4c). Taken together, three genetic networks reconstructed from SNP data can be reasonably well interpreted by annotated protein-protein interaction networks, showing the biological relevance of our model.

Beyond most existing undirected networks, our networks can characterize causal epistasis. In the third-layer network of SM17, QTL Q2728755 (located at *SAOUHSC_02967*) establishes an "antagonistic" relationship with SNP S2748895 (*SAOUHSC_02982*); i.e., while this QTL inhibits the latter, it is also inhibited by the latter (Fig. 4b). Meanwhile, this QTL receives strong promotion from S2773077(*SAOUHSC_03000*), S2773290 (*SAOUHSC_03000*), and S2085718 (*SAOUHSC_02250*). Together, although Q2728755 has a small independent effect, i.e., a limited intrinsic capacity to express itself in isolation, its net genetic effect detected by coFunMap is still remarkable, thanks to large favorable contributions from multiple positive regulators (Fig. 4d). Thus, by activating the expression of these three regulators and/or inhibiting the expression of S2748895, the genetic effect of Q2728755 on phenotypic plasticity can be altered. SNP S108000 is an insignificant SNP by coFunMap, but it is in close proximity to gene *SAOUHSC_00103* that codes phosphonates ABC transporter permeases of high relevance to antibiotic resistance[81] and, thereby, is expected to exert a large independent effect. The net effect of this SNP does not achieve a significance level because inhibition it receives from two physically close suppressors (S2773077 and S2773198) is stronger than the promotion it receives from a promoter (S2748895).

The third-layer networks of submodules SM29 and SM25 each contains two QTLs (Fig. 4b). In each case, we find that the significance of these QTLs is largely contributed by favorable regulation from other SNPs. In submodule SM29, Q768340 (located at *SAOUHSC_00786*) has a large independent effect that is amplified through the promotion of Q229384 (in a non-coding region) at the early stage of growth, leading to a larger net genetic effect (Fig. 4d). Q229384 is also promoted by Q768340, both of which thus form a mutualistic relationship. Because these two QTLs are not regulated by any other SNP, they can be genetically manipulated as a cohesive unit to alter phenotypic plasticity. As expected, both Q2815318 and Q2072412 in submodule SM25 display a subtle independent effect (Fig. 4d) because they are located in a non-coding region (Table S2). These two QTLs form an "amensalistic" relationship; i.e., the former inhibits the latter whereas the latter is neutral to the former. Despite the inhibition of Q2815318, Q2072412 receives, to a larger extent, promotion by S319914 (*SAOUHSC_00304*), making Q2072412 a significant net genetic effect.

Like the case of module M12, as described above, in the second-layer submodule networks of modules M14 (Fig. S3A), M7 (Fig. S4A), and M10 (Fig. S5A), all QTL-containing submodules receive more incoming links than send outgoing links. It appears that positive regulation is more frequent than negative regulation in the second-layer network of M14, whereas the inverse pattern is found for the second-layer networks of M7 and M10. We reconstruct the third-layer networks at the SNP level for QTL-containing submodules of modules M14 (Fig. S3B), M7 (Fig. S4B), and M10 (Fig. S5B), from which we can clearly investigate how each QTL interacts with other SNPs. Q550312 (located at gene *SAOUHSC_00544*) in the submodule SM25 of M14 (Fig. S3C) has a subtle independent effect at the early stage of growth, which yet increases dramatically at the middle stage. However, the net effect of this QTL is strikingly large during the entire growth because it receives a favorable promotion from S1333141 (in a non-coding region) and S2818604 (located at gene

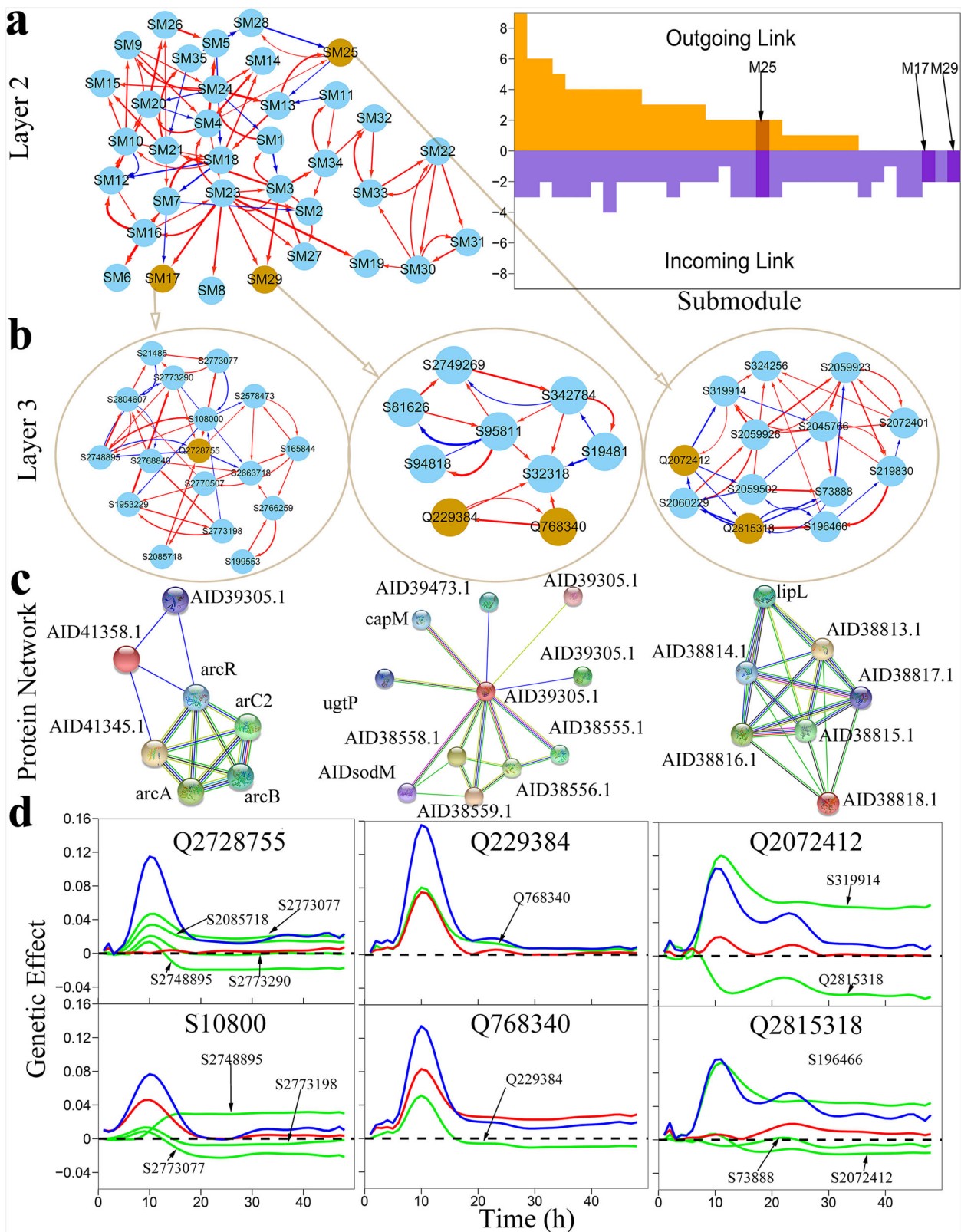

*gidA*). *gidA* is found to play a role in regulating the expression of *salA* associated with lantibiotic production in bacteria [82,83].

A total of four and three QTLs are detected in the submodules SM24 and SM25 of M7, respectively (Fig. S4B). These QTLs perform differently in affecting phenotypic plasticity. Some QTLs, such as Q25261 (at *SAOUHSC_00020*), Q193712 (at *SAOUHSC_00176*), Q92210 (at *SAOUHSC_00085*), and Q2644158 (at *SAOUHSC_02871*), have significant net effects mainly due to favorable accumulated promotion from regulators, although the strength of promotion and number of regulators differ from QTL to QTL (Fig. S4C). For Q550323 (at *SAOUHSC_00544*), its independent effect is even larger than its net effect because it receives a negative regulation from S119306 (in a non-coding region). As has been seen from the above gene

**Fig. 4 The second- and third-layer network of QTL-containing module M12. a** Left panel: The 35-node inter-submodule network at the second layer under network community M12 of the first-layer network. Submodules SM17, SM25, and SM29 contain QTLs, highlighted in orange. Red and blue arrowed lines stand for the direction of up-regulation and down-regulation, respectively, with the thickness of lines proportional to the strength of regulation. Right panel: Distribution of the number of outgoing and incoming links over 35 submodules. The positions of QTL-containing submodules are indicated. **b** The 17-node, 9-node, and 13-node inter-SNP network at the third later under QTL-containing network subcommunities SM17, SM29, and SM25, respectively, of the second-layer network M12. QTLs are highlighted in orange. **c** Interaction networks among proteins (with names indicated) annotated from SNP networks, where the strength of protein interaction is proportional to the thickness of linked line. **d** Curves of genetic effects on phenotypic plasticity for QTLs from SM17, SM29, and SM25. For comparison, an insignificant SNP108000 from SM17 is indicated. Net genetic effect (blue line) is decomposed into independent effect (red line) and dependent effects (green line) due to regulation by other SNPs.

enrichment analysis (Table S2), Q550323 is just located at gene *SdrC* that is directly involved in vancomycin resistance[63,64], thus explaining why this QTL has a larger independent effect. Yet, Q448470, located in a non-coding region (Table S2), has a sizeable independent effect (Fig. S4C), a phenomenon deserving a further investigation. Three QTLs are detected in module M10 (Fig. S5B), whose net, independent and dependent effects on phenotypic plasticity can be dissected in a similar way (Fig. S5C). It should be pointed out that Q183488 has a large independent effect during the most time of growth, which may be due to the fact that it is within gene *SAOUHSC_00169* (*oppA*) encoding peptide ABC transporter permeases (Table S1), critical for antibiotic resistance[84].

**Retrieving missing heritability from multilayer networks**. Traditional computational strategies only detected 16 QTLs from 25,173 SNPs, which may not fully explain the whole genetic variation of phenotypic plasticity. One common approach for detecting missing heritability is to produce more SNPs throughout the genome using deeper sequencing or other genetic variants, such as epigenetic marks. However, the efficiency of such an approach still depends on the way the data are analyzed. We argue that some insignificant SNPs undetected by functional mapping may be potentially used to retrieve missing heritability through genetic networks. As an example, we choose a non-QTL-containing module M4 to dissect how its SNPs trigger genetic effects on phenotypic plasticity. We break down this module composed of 5,419 SNPs into 464 submodules and further reconstruct a 464-node second-layer network at the submodule level (Fig. 5a). Both outgoing and incoming links constituting this submodule network are scale-free, with the frequency of links following an exponential function. Yet, these two types of links differ dramatically in many aspects. Outgoing links are much more numerous than incoming links, but the former is overwhelmingly predominated by only a few leaders, whereas the latter is relatively evenly distributed. Taken together, in a large genetic network, a small portion of genetic loci play a leadership role in maintaining network behavior, and almost all loci tend to receive regulation from one or more other loci.

As a popular regulator, submodule SM437 (containing 24 SNPs) in the M4 network influences many other submodules (Fig. 5a). We reconstruct a 24-node third-layer genetic network for SM437 at the SNP level (Fig. 5b). Pathway analysis using STRING[80] identifies that two linked SNPs S964411 (residing within 2000 bp from gene *SAOUHSC_00991*) and S964453 (located at *SAOUHSC_00992*) code two interacting proteins AID39489.1 and AID39490.1 (Fig. 5c), suggesting the biological relevance of the SNP network. We randomly choose five insignificant SNPs detected to be located at the bottom of the Manhattan plot (Fig. 2b) for their effect decomposition. As shown in Fig. 5d, S366529 (in a non-coding region) and S964411 (at gene *SAOUHSC_00994*), although found to be insignificant by functional mapping, display large independent genetic effects on phenotypic plasticity. The former is inhibited by two SNPs,

whereas the latter is promoted by one SNP but inhibited, to a much larger extent, by a second SNP. The negative regulation they receive from other SNPs cancels out their independent effect, making their net genetic effects insignificant. The other three SNPs have consistently small independent effects, but their small net effects undergo different mechanisms. S855041 (at *SAOUHSC_00891*) receives a strong positive regulation and, in the meanwhile, a strong negative regulation, which cancels each other so that its net effect makes no change. S395434 (at *SAOUHSC_00392*) and S678553 (at *SAOUHSC_00693*) each receive a small regulation from one SNP, which does not influence their independent effects. Taken together, although many SNPs are detected to be insignificant by traditional approaches, they may not be necessarily unimportant rather they are intrinsically impactful in nature (like S366529 and S964411). Even for those intrinsically insignificant SNPs (such as S855041), their net effect can be changed through altering the expression of its regulators. These results suggest a possibility to retrieve missing heritability hidden in genetic architecture by editing the relevant regulators that act through genetic networks.

**Multilayer interactome networks underlying biotic phenotypic plasticity**. We used the same procedure to map the genetic architecture of biotic phenotypic plasticity by incubating a panel of *S. aureus* strains in monoculture and co-culture with *E. coli*. In general, *S. aureus* strains display a reduced growth from monoculture to co-culture[60], showing remarkable plastic response to species coexistence (Fig. S6A). We implement coFunMap to identify eight QTLs for biotic phenotypic plasticity from 12,849 SNPs (Fig. S6B), each exhibiting different temporal patterns of genetic effects over time (Fig. S6C). Gene annotations of these QTLs show their unique biological functions (Table S3). Based on the similarity of genetic effect trajectories, coFunClu classifies 12,849 SNPs into nine distinct modules (Fig. S7A), with each module representing a different network community within a whole network. All eight QTLs are located in two modules M1 (2) and M3 (6). GO analysis shows that SNPs in different modules reside in the genomic regions of different genes (Fig. S7B). While modules M4 and M6 includes genes that encode a wide range of biological functions, QTL-containing modules are biologically very specific, as observed in the first GWAS study (Fig. 3b). Genes in M1 code biosynthesis of amnio acids, lysine biosynthesis, valine, leucine and isoleucine degradation, and biosynthesis of various secondary metabolites, whereas genes in M3 are associated with *S. aureus* infection, glycerolipid metabolism, and O-antigen nucleotides sugar biosynthesis.

We reconstruct a multilayer interactome network that covers all 12,849 SNPs for *S. aureus*' phenotypic plasticity to co-existing *E. coli*. At the top layer (layer 1) is a 9-node inter-module network (Fig. 6a), in which each module is a network community and one module is linked to the next through bDSW interactions. For example, M2 establishes an antagonism relationship with M4, but a mutualism relationship with M5. M6 is "parasitic" to M1 by inhibiting the promoting latter and, meanwhile, is "altruistic" to

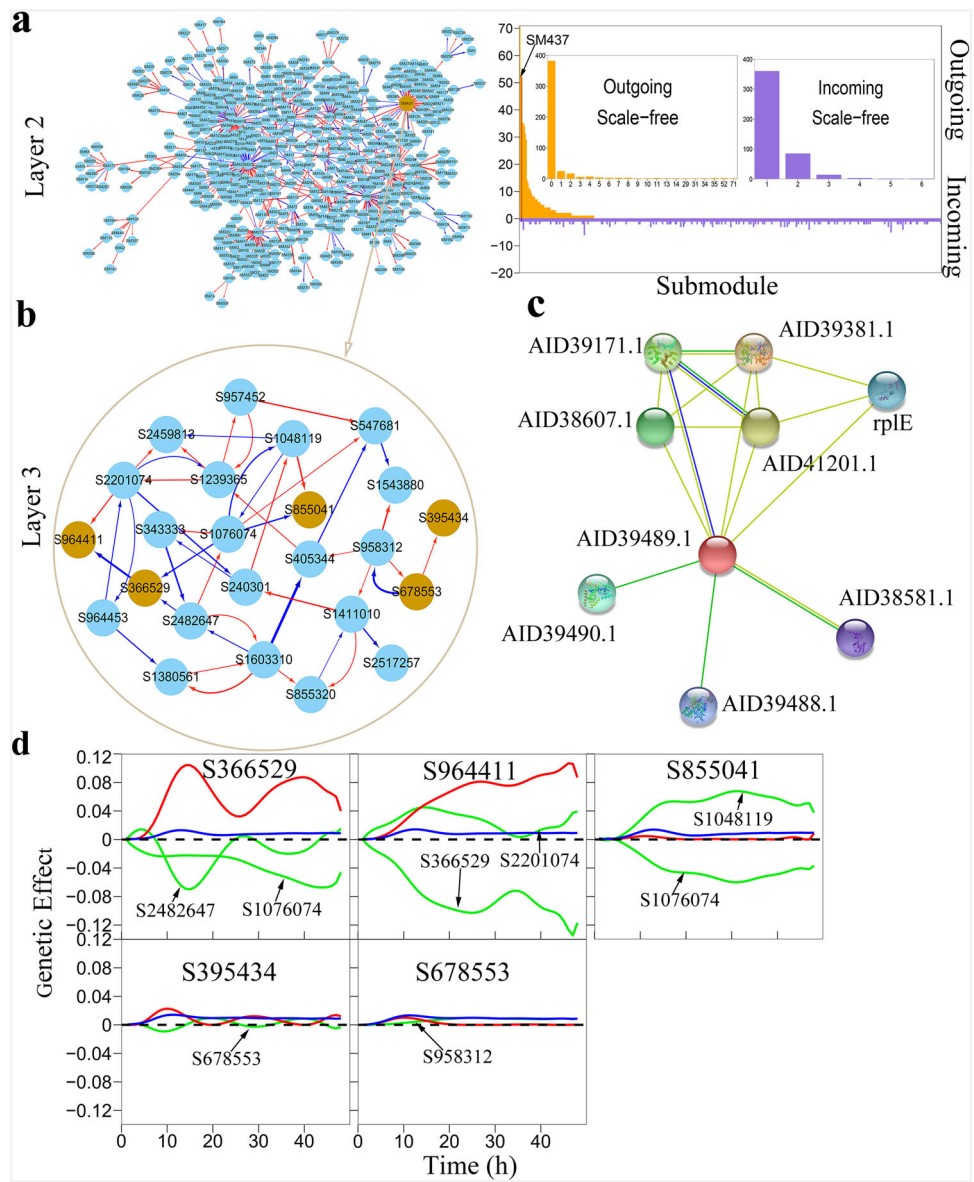

**Fig. 5 The second- and third-layer networks of module M4 containing no QTL. a** Left panel: The 464-node inter-submodule network at the second layer under network community M4 of the first-layer network. Right panel: Distribution of the number of outgoing and incoming links over 464 submodules whereas one of the leaders SM437 is indicated. Within the distribution plot, two smaller plots show the power distribution of the frequency of outgoing and incoming links. **b** The 24-node inter-SNP network at the third later under leader submodule SM437 of the second-layer network M4. **c** Interaction network among proteins (with names indicated) annotated from SNP network, where the strength of protein interaction is proportional to the thickness of linked line. **d** Curves of genetic effects on phenotypic plasticity for insignificant SNPs chosen from the third-layer network in (**b**). Net genetic effect (blue line) is decomposed into independent effect (red line) and dependent effects (green line) due to regulation by other SNPs.

M9 by promoting the inhibiting latter. M4 and M6 with more outgoing links than other modules are regarded as leaders. QTL-containing modules M1 and M3 tend to receive incoming links, suggesting that these modules contain target genes. As seen from the above GO analysis (Fig. 3b), the two leaders cover more biological functions than target modules. All these findings are observed to be consistent with those detected in the abiotic GWAS experiment.

We implement coFunClu to further classify each module into its constituent submodules and reconstruct the second-layer networks at the submodule level. Figure 6 elucidates three such submodule networks for two QTL-containing modules M1 and M3 and a leader module M6. Unlike a case for the first-layer network, promotion tends to occur at a greater frequency than inhibition within the second-layer networks. For a big

submodule, we further classify it into sub-submodules to reconstruct the third-layer networks, but for a small submodule, we directly reconstruct SNP-SNP networks at the bottom layer. We show SNP networks for QTL-containing submodules in which the roadmap of how a QTL interacts with other loci toward phenotypic plasticity can be charted (Fig. 6a). As a comparison, we also provide a SNP network for a representative QTL-absent submodule. We perform a pathways analysis for SNP networks, suggesting that SNP-SNP interactions detected by our model can be annotated to protein-protein interactions (Fig. 6b). In the SM4 network of QTL-containing M1, two interacting SNPs S1464883 (at *SAOUHSC_01515*) and S1464685 (at *SAOUHSC_01516*) code proteins AID38810.1 and AID38809.1, respectively, which are linked. A similar finding is true for another QTL-containing module M4; i.e., AID39944.1-AID39945.1 link interpreted by

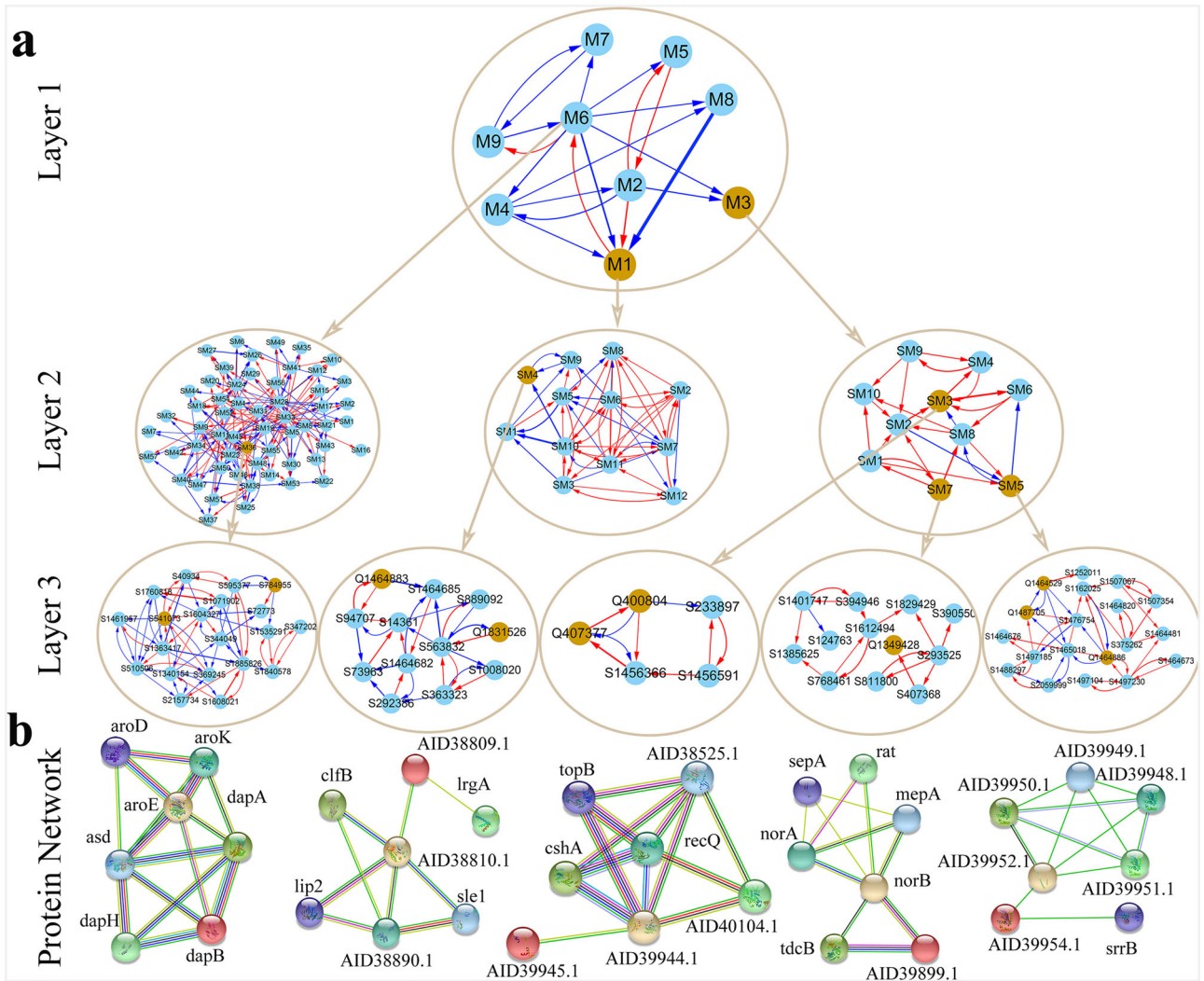

**Fig. 6 Multilayer genetic networks reconstructed from a high-dimensional genome-wide SNP data in the biotic GWAS experiment. a** On the top is the nine-node network at the first layer among nine modules each presenting a network community. Under the first-layer network are the networks at the second layer reconstructed from submodules of leader module M6 and QTL-containing modules M1 and M3 (highlighted). The second-layer networks go to the third-layer networks at the SNP level from particular submodules. **b** Interaction networks among proteins (with names indicated) annotated from SNP networks, where the strength of protein interaction is proportional to the thickness of linked line.

S1456366 (at *SAOUHSC_01502*)-S1456591 (at *SAOUHSC_01503*) interaction in the SM3 network, norB-AID39899.1 link by S1385625 (at *SAOUHSC_01448*)-S1456591 (at *SAOUHSC_01450*) interaction in the SM7 network, and AID39952.1-AID39952.1 link by S1507067 (at *SAOUHSC_01580*)-S1507354 (at *SAOUHSC_01582*) interaction in the SM5 network. Even for modules without QTLs, SNP-SNP interactions detected can correspond to protein-protein interactions. For example, in the SM437 network of M4, how S964411 (at *SAOUHSC_00991*) interacts with S64453 (at *SAOUHSC_00992*) determines the link of their coding proteins AID39489.1 and AID39490.1 (Fig. 6b).

Figure 7 provides a detailed atlas through which a QTL affects the plastic response of *S. aureus* to species coexistence. For some QTLs, such as Q400804 (locate at *SAOUHSC_00397*), Q407377 (at *SAOUHSC_00405*), Q1349428 (in a non-coding region), and Q1464883 (at *SAOUHSC_01515*), observed genetic effects are larger than their intrinsic independent effects due to up-regulation they receive from regulators. This is especially true for Q1349428 with an almost neglected independent effect, whose significance is derived from a tremendous dependent effect it receives from S407368 (at *SAOUHSC_00405*). This finding

suggests that the practical transformation of Q1349428 can only be made possible through a joint selection or editing of S407368. Q1831526 (at *SAOUHSC_01926*) is significant by coFunMap, but its independent effect is even larger than what can be observed. Down-regulation this QTL receives from S563832 (in a non-coding region) cancels its intrinsic capacity to affect phenotypic plasticity. Thus, by silencing the expression of S563832, the effect of Q1831526 can be better released.

In the preceding section, we argue that our network can potentially detect missing heritability. This argument is confirmed by our biotic GWAS study. We randomly chose two insignificant SNPs to draw their effect curves (Fig. 6). Although they are insignificant, the mechanisms underlying their effects differ dramatically. S541073 (located ta *SAOUHSC_00535*) has a subtle independent effect. Because it only receives a subtle dependent effect, its net effect is very small. Thus, this SNP makes no contribution to phenotypic plasticity. However, a dramatically different pattern is observed for S784955 (at *SAOUHSC_00802* encoding carboxylesterase). This SNP has a formidable independent effect, but because of striking down-regulation by SNP595377 (in a non-coding region) its observed effect almost

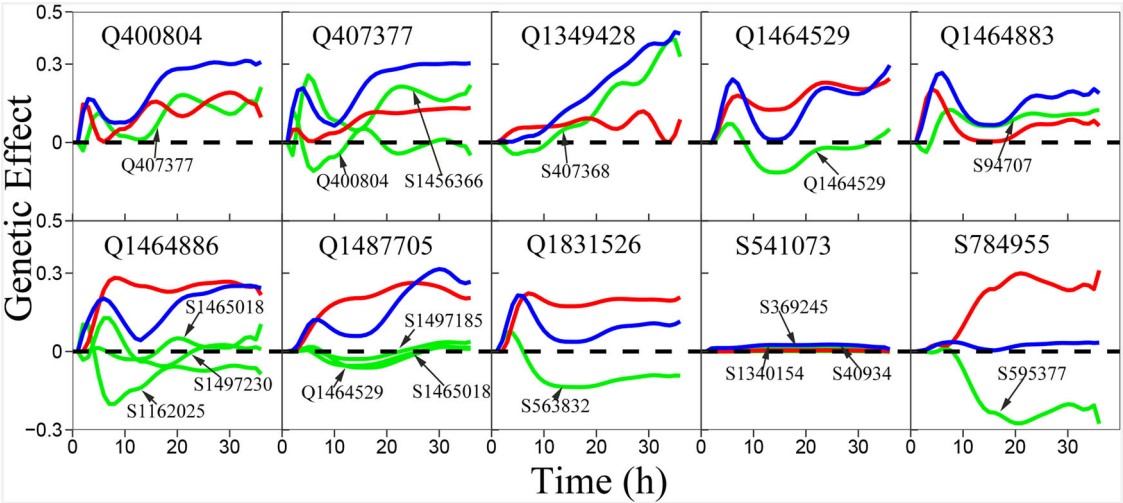

**Fig. 7 Genetic effect curves of eight QTLs and two randomly chosen insignificant SNPs.** Net genetic effects for *S. aureus'* response to *E. coli* coexistence in the biotic GWAS experiment (blue line) are decomposed into independent effects (red line) and dependent effects (green line) due to regulation by other SNPs.

can be neglected. In practice, the real effect of S784955 on biotic defense can be released if its negative regulator is knocked out. We argue that those SNPs, like S784955, can be used to retrieve missing heritability.

**Comparison with polygenic models**. Linear mixed models (LMM) are one of the most commonly used computational approaches for GWAS[85]. The advantage of LMM lies in its capacity to correct population structure and kinships so that genetic architecture can be more precisely characterized. Our network model is qualitatively different from LMM in two key aspects. First, LMM can only analyze static phenotypes measured at a single time point, whereas our model makes use of dynamic phenotypes that are viewed as curves. Second, most LMM attempt to detect single significant loci at a time from a pool of genome-wide SNPs, but our model simultaneously analyzes all SNPs from which all possible pairwise interactions are characterized. Third, LMM assumes a linear genotype-phenotype relationship, whereas our model can capture any nonlinear relationship. Despite these differences, we can still make a rough comparison between the two models. We first reduce the growth dimension from dynamic to static by calculating growth rates for each strain and implement an LMM to estimate the genetic control of growth rate under each condition. In the abiotic phenotypic plasticity experiment of *S. aureus*, the total heritability of growth rate explained by all QTLs is 0.516 in vancomycin-free condition and 0.021 in vancomycin-exposed condition. By combining data from two conditions, we estimate the overall heritability of growth rates (explained by all genes in the genome) as 0.763, larger than the total QTL heritability, suggesting the existence of tremendous missing heritability. Similarly, we identify tremendous missing heritability in the biotic phenotypic plasticity experiment, where the overall heritability of growth rate is 0.602, whereas the total heritability explained by all QTLs is 0.170 in monoculture and 0.335 in co-culture. Taken together, although LLM can precisely identify individual QTLs, they may be encountered with a formidable issue of missing heritability. Because of a joint analysis of all SNPs and their pairwise interactions, our model show its power to detect missing heritability. Furthermore, our model can provide the roadmap of gene regulation, helping geneticists to retrieve and excavate some hidden missing heritability by blocking negative regulation.

## Discussion

Phenotypic plasticity involves a complex genetic component[8–12], but its genetic analysis is based on reductionist thinking of individual gene identification. As a composite trait derived multiple phenotypes expressed in different environments, phenotypic plasticity is likely to comply with omnigenic theory[30]; i.e., it may be controlled by all genes an organism may carry. By integrating mapping theory and elements of multiple disciplines, we develop a computational model to encapsulates all SNPs from a GWAS into a phenotypic plasticity-driven genetic network. Different from gene regulatory networks from expression data[86,87], our genetic networks unravel a direct link from DNA genotype to phenotype, providing a broader picture of genetic architecture.

One major advantage of our model lies in its capacity to chart a comprehensive roadmap of how each SNP flows its genetic signal towards phenotypic plasticity, whether a SNP exerts its effect either in its own capacity (independent effect) or through regulation by promotors or inhibitors (dependent effect), or both, and how the intrinsic effect of a SNP is masked by other loci. To validate the biological relevance of our model, we design and conduct two independent GWAS experiments by culturing each panel of *S. aureus* strains in two contrast environments. These two experiments are complementary, one focusing on the genetic mapping of phenotypic plasticity to an abiotic environment (vancomycin-free vs. vancomycin-exposed) and the second on the genetic mapping of phenotypic plasticity to a biotic environment (isolation vs. socialization). We use our model to reconstruct a large-scale, multilayer, and maximally informative genetic network underlying abiotic phenotypic plasticity and biotic phenotypic plasticity, respectively. The two networks unravel many previously undetected genetic mechanisms by traditional mapping approaches and, also, produce findings that are reciprocally supported from the two GWAS experiments.

From the genetic networks reconstructed from two GWAS experiments, we identify common causes of the significance of QTLs detected by genetic mapping based on marginal effects of single SNPs. coFunMap finds 16 and eight key QTLs of significant marginal effects that mediate *S. aureus'* response to vancomycin and to *E. coli* coexistence, respectively, in the abiotic and biotic GWAS. We postulate three mechanisms underlying the significance of these QTLs.

A QTL is significant mainly because of its intrinsic capacity to express a strong independent genetic effect. Q550323 is a QTL

detected in the abiotic GWAS experiment. Its tremendous independent effect on vancomycin susceptibility estimated by our model (Fig. S4C) is in good agreement with the direct role of gene *SAOUHSC_00544* (*SdrC*) in mediating vancomycin resistance[63,64] at which this QTL is located (Table S2; Fig. S8). Q550323 is so highly expressed in its own capacity that its net effect is still remarkably large even if it receives a negative regulation from S119306 (in a non-coding region). Q1831526 is a QTL located at gene *SAOUHSC_01926* coding a hypothetical protein (Table S3), which is detected to affect the plastic response of *S. aureus* to the coexistence of *E. coli* in the biotic GWAS experiment (Fig. 7). This QTL displays a sizeable independent effect, which can be observed (net effect) even if it is down-regulated by a negative regulator S563832 (in a non-coding region). In practice, by silencing the expression of S119306 and S563832 vis gene editing, the effect of Q550323 on abiotic phenotypic plasticity and the effect of Q1831526 on biotic phenotypic plasticity can be amplified, respectively.

Most of the QTLs detected from both GWAS experiments affect phenotypic plasticity through this mechanism, i.e., making use of favorable regulation from other regulatory genes rather than their own intrinsic capacity. For example, Q2728755 located at *SAOUHSC_02967* of the *S. aureus* genome (detected from the abiotic GWAS experiment) does not affect vancomycin susceptibility in its own intrinsic capacity, but it does exert a remarkable effect, especially in the early stage of vancomycin susceptibility, through positive regulation from S2085718 (at the *SAOUHSC_02250* location), S2773077 and S2773290 (both at the *SAOUHSC_03000* location) (Figs. 4c, S8). *SAOUHSC_03000* (*capA*) gene is involved in the biosynthesis of outer capsules of cell walls[88], which provides material support for Q2728755 to mediate *S. aureus* colonization, pathogenesis, and bacterial evasion of the host immune defenses. The net effect of Q2728755 decreases considerably with time because it receives negative regulation by S2748895 (at *SAOUHSC_02982*). Thus, by inhibiting the expression of S2748895 and/or promoting the expression of S2085718, S2773077, and S2773290, it is possible to amplify and maintain the genetic effect of Q2728755 in response to vancomycin.

The expression of Q193712 detected in the abiotic GWAS experiment is promoted jointly by three SNPs and also inhibited, but to a lesser extent, by one SNP (Figs. S4C, S6), leading the net effect of this QTL to be larger than its independent effect. One of the promotors is S550323 (at *SAOUHSC_00544*) that is proximal to the fibrinogen-binding protein *SdrD* gene encoding serine-aspartate repeat-containing protein D. This protein can promote the adhesion of bacteria to host cells, help resist the killing of innate immune components, such as neutrophils in the blood, and, thus, weaken bacterial clearance[89]. Q2783126 (detected from the abiotic GWAS experiment), located at gene *SAOUHSC_03008* coding multiple phosphatase-related processes, is not expressed in its own capacity when its growth is at a linear stage (Fig. S4C; Fig. S8), but because of accumulative up-regulation from many SNPs, it displays a tremendous net effect. When entering a stationary growth stage, Q2783126 is largely inhibited by S220967 (at *SAOUHSC_00199*) and S183439 (at *SAOUHSC_00169*), making its net effect below its increasing independent effect. As an inhibitor, *SAOUHSC_00169* encodes ABC transporter permease that reduces intracellular antibiotic concentration through MDR proteins[90,91], thereby leading to change in drug resistance.

Some QTLs are significant because of both intrinsic capacity and extrinsic regulation. For example, Q183488 detected from the abiotic GWAS experiment exerts a large independent effect on vancomycin susceptibility (Fig. S5C) probably because of its proximity to gene *SAOUHSC_00169* (*oppA*) that plays a critical role in antibiotic resistance through encoding peptide ABC

transporter permeases[83] (Table S2). This QTL receives a mix of positive and negative regulation from three SNPs, making its net effect change with time in a cyclic pattern. SNP S213345, located at *SAOUHSC_00192* encoding coagulase, promotes or inhibits the effect of Q183488, depending on stages of vancomycin response. Staphylocoagulase (*Coa*) is involved in the formation of biofilms[92] and can promote blood coagulation by activating prothrombin to convert fibrinogen into fibrin[93]. By changing the expression of regulators, the role of Q183488 in mediating vancomycin resistance can be maximized.

QTL400804 detected from the biotic GWAS experiment is a QTL locate at gene *SAOUHSC_00397* encoding type I restriction-modification system subunit M (Table S3). This QTL has a large independent genetic effect on *S. aureus*' response to the coexistence of *E. coli*, which is amplified by its altruistic Q407377 in a proximity to gene *SAOUHSC_00405* encoding a hypothetical protein (Table S3; Fig. 7). Q407377 is strongly expressed in its own capacity and, meanwhile, receives up-regulation from S1456366, leading to its increasing net genetic effect, despite a negative regulation by the parasitic QTL400804.

We provide a complete picture of how the QTLs detected by traditional approaches affect phenotypic plasticity directly or through indirect regulation from other genes. In each mechanism described above, the pattern, strength, sign, and number of regulations differ from QTL to QTL. A detailed understanding of these mechanisms is of great help to best utilize these QTLs for improving phenotypic plasticity, i.e., drug resistance or species interaction as described in this study, through marker-assisted selection or gene editing.

In the past decades, GWAS has been criticized on the identification of only a small portion of genetic variance, leaving its main part as missing heritability[94,95]. There have been a body of literature on retrieving missing heritability by finding new genetic variants[96]. We can methodologically explore this issue through multilayer interactome networks. By dissecting the net genetic effect of these SNPs into their independent and dependent effects, our model characterizes novel interpretations. For example, S964411, located at gene *SAOUHSC_00994*, codes bifunctional autolysin (Fig. S9), but it is detected to be insignificant for vancomycin susceptibility by coFunMap (Fig. 5c). However, this SNP would display a pronounced effect if it was in a socially isolated circumstance, because its insignificance is largely due to negative regulation by a non-coding S2201074. Although S964411 also receives positive regulation from a third SNP, this regulation does not adequately cancel the negative regulation. Thus, through inhibiting the expression of regulator S2201074, we expect that S964411 can fully execute its intrinsic capacity to deliver an impactful effect on *S. aureus*' response to vancomycin. Similarly, S784955 located at *SAOUHSC_00802* encoding carboxylesterase is observed to be insignificant for *S. aureus*' reaction to E. coli, but it displays a large independent effect. Thus, if we silence the expression of its inhibitor, non-coding SNP595377, S784955 can greatly contribute to biotic defense. Taken together, those insignificant SNPs either with strong independent effects or regulated by promotors or inhibitors, or both, can be used to retrieve a certain amount of missing heritability is hidden in regulatory networks by uncoupling unfavorable regulation and/or strengthening favorable regulation.

Despite its importance for trait control, epistasis is difficult to estimate without a considerable large sample size that is often hardly met in practice. For example, Zuk et al[34]. suggested that as many as 10,000 samples are required to estimate a genetic interaction in GWAS. However, as have been shown above, the characterization of dependent effects is not dependent on sample size but on net genetic effect curves. The estimation of main genetic effect curves does rely on sample size, but both simulation

and experimental studies suggest that a sample size of 100 to 400 can reasonably well estimate such curves if the number of time points is two or more times the number of curve parameters[97,98]. Furthermore, traditionally defined epistasis is the genetic interaction between different loci, without knowing the direction and sign of epistasis. Our model can capture the full properties of epistasis, i.e., strength, signed, and direction by estimating dependent effects in our ODE model (equation (9)). Taken together, our model can detect any epistasis and identify its mechanism from any number of SNPs from a GWAS, but much less reliant upon sample size compared to traditional approaches.

Our model integrates two emerging theories in quantitative genetics, i.e., omnigenic control and nonlinear gene–environment interactions[32,39]. By borrowing ingredients of various disciplines, these two theories are coalesced into a unified framework by which the genetic architecture of phenotypic variation and plasticity can be systemically characterized. However, our model has several limitations. First, it needs dynamic data of traits, which may not be widely available in ordinary general GWAS. Yet, Wu and Jiang's[50] strategy for extracting dynamic information from static snapshots may be incorporated into our model to expand its use to static phenotypic data. Second, our model can only be used for a single phenotype, but the organism can be better described by its multifaced correlated features. However, how to reconstruct genetic networks for the phenotypic plasticity of multiple phenotypes is not a simple extension of the current model because these phenotypes may be interdependent of each other in a complex, likely nonlinear, manner. Third, our model only considers two environments. To extract the complete genetic mechanisms underlying how complex traits function, vary, and evolve along a spatiotemporal gradient, a series of environments should be sampled, but joint modeling of multiple environments cannot be made possible without the implementation of sophisticated spatial statistical methods. Despite these areas remaining to be explored, the precise mapping of SNP-SNP interactions identified by our model to the interactions between coding proteins, confirmed with two independent S. aureus GWAS experiments, implies its biological relevance and potential promise to be used as a general quantitative genetic approach at a new but higher standard. If this is a case, our model can likely make a paradigm shift in trait mapping from reductionist thinking of individual genes to a holistic approach that portrays a complete picture of genetic mechanisms.

## Methods

The motivation of this study is to develop a computational model for mapping the complete genetic architecture of phenotypic plasticity in a GWAS. To better describe the rationale and derivation procedure of the model, we start with two real GWAS experiments for an important bacterial species S. aureus. These two experiments aim to address the same biological question about phenotypic plasticity from different perspectives. The first experiment is designed to study how S. aureus responds to an abiotic environment, whereas the second experiment attempts to answer the question of how this species alters its phenotype in response to a biotic environment.

**Abiotic mapping experiment**. We sampled and cultured 99 S. aureus strains (with serial numbers given in Table S1) in both vancomycin-free medium (control) and vancomycin-exposed (6 μg/mL) medium (stress). All S. aureus strains (numbered as S1–S9, S11–S36, S40–S45, S47–S85, S87–S105), of which 41 are vancomycin-resistant to different extents and 58 are vancomycin-sensitive, were obtained from parental strains via a 60-day in vitro vancomycin treatment. All original parental vancomycin-sensitive strains (numbered as $S_p1$–$S_p9$, $S_p11$–$S_p36$, $S_p40$–$S_p45$, $S_p47$–$S_p85$, $S_p87$–$S_p105$) are VSSA (MIC = 1–1.5) (Table S1), collected from China Industrial Microbial Strain Preservation Center, China Agricultural Strain Preservation Center, China Forestry Microbial strain Preservation and Management Center, China General Microbial Strain Preservation Center, China Pharmaceutical Microbial Strain Preservation and Management Center, Chinese Typical Culture Preservation Center, China Medical Microbial Strain Preservation and Management Center, China Agricultural University, and Beijing Chaoyang Hospital (Table S2). The strains were stored in cryogenic refrigerator at −80 °C. Each

parental strain was inoculated on BHI agar plate containing 1/2 initial MIC vancomycin and transferred to fresh medium with the same vancomycin concentration every 24 h for 4 days. Strains MICs were re-determined every 4-day and the treatment repeated with updated MIC. The MICs of all strains (except for S10) were elevated in different degrees after 60 days in vitro treatment (Table S1).

To obtain growth data, we inoculated each strain in the same-volume liquid medium under control and stress and measured their $OD_{600}$ to represent growth abundance at 1 h after culture, once every 2 h after 2 h till 12 h, once every 4 h after 12 h till 24 h, and once every 6 h after 24 h till 48 h. In total, there are 14 time points for measurement. Genomic DNA of strains was extracted by TIANamp Bacteria DNA Kit (TIANGEN, Beijing, China) according to the manufacturer's protocol. Whole-genome sequencing was performed using Illumina HiSeq 4000 instrument (Illumina Inc., San Diego, CA, United States) at Allwegene (Beijing, China). To obtain initial alignment results, sequencing data were compared with the reference genome (S. aureus subsp. aureus NCTC 8325) using BWA mapper v0.7.8[99]. SAMtools v0.1.18[100] was used to sort the alignment results, producing 110,678 SNPs, of which 25,173 pass quality control for association mapping. All sequence data could access in the Sequence Read Archive (SRA) data NCBI (https://www.ncbi.nlm.nih.gov/Traces/study/?acc=PRJNA722566).

**Biotic mapping experiment**. We collected 100 S. aureus strains and 100 E. coli strains from the National Infrastructure of Microbial Resources, China. We pre-cultivate these microbial strains using a procedure as described previously[60,101]. After pre-cultivation, we inoculated 100 strains of S. aureus (monoculture) and 100 independent S. aureus–E. coli interspecific strain pairs (co-culture) in different flasks filled with the same media, i.e., 25 mL two-time diluted brain heart infusion medium (OXOID, Basingstoke, England). For co-culture, we mix two species in a 1:1 ratio. All the flasks were incubated at 30 °C and shaken at 130 rpm. To minimize the position effect, we placed the flasks in a randomized layout. The experiment was repeated three times.

Each strain in a flask was sampled at time once every 0.5 h before 2 h of culture, followed by once every 2 h after 2 h and once every 4 h after 12 h, with a total culture period of 36 h. Quantitative PCR (qPCR) measurements of S. aureus were performed according to the previously reported procedure[60]. The qPCR counts of each strain, averaged over three replicates, are used as abundance phenotypes for genetic mapping. Whole-genome sequencing of 100 E. coli strains was performed on the Illumina HiSeq platform at Allwegene (Allwegene, Beijing, China). In total, we obtain 12,849 SNPs for S. aureus. The resequencing statistics for the genomes of S. aureus are summarized in Tables S4.

**Growth equation fitting**. Changes of abundance with time are important phenotypes that describe the biological properties of microbes. Time-varying abundance of a strain in both biotic and abiotic GWAS experiments can be fitted by growth equations. A number of growth equations have been developed, among which Gompertz, logistic, and Richards equations are most commonly used to fit microbial growth[102,103]. In general, these equations produce biologically similar results[104], from which we choose the logistic equation to fit our data, expressed as

$$g(t) = a\left[1 + b \cdot \exp(-rt)\right]^{-1}, \qquad (1)$$

where $a$ is the asymptotic growth of the strain, $b$ is the parameter that describes the initial growth of the strain, and $r$ is the average specific growth rate of the curve. To characterize the growth properties of an organism, Sun et al[105]. proposed to use several key heterochronic parameters that can specify the timing, period, rate, and event of development. Two of the heterochronic parameters, specific growth rate ($r$) and the timing of inflection point ($t_I$) at which the strain reaches a maximum relative growth rate, calculated as $t_I = (\ln b)/r$, were used to compare growth curves.

**Quantifying phenotypic plasticity**. Consider a GWAS population of $n$ members, i.e., strain, as described above, genotyped by $p$ genome-wide SNPs. To study the phenotypic plasticity of microbial abundance, we culture each strain in two contrast environments and measure its abundance repeatedly at a series of $T$ time points after culture. Let $\mathbf{x}_i = (x_i(t_1), \ldots, x_i(t_T))$ and $\mathbf{y}_i = (y_i(t_1), \ldots, y_i(t_T))$ denote a vector of abundance values measured for the $i$th strain at time schedule $(t_1, \ldots, t_T)$ in environment X and Y, respectively. The time-dependent differences of microbial abundance for the $i$th strain between two environments is calculated as

$$\mathbf{z}_i = (z_i(t_1), \ldots, z_i(t_T)) \equiv ((x_i(t_1) - y_i(t_1)), \ldots, (x_i(t_T) - y_i(t_T))), \qquad (2)$$

where the size and sign of each element at a specific time point reflect the degree and pattern of how strain $i$ responds to environmental change from X to Y over the time course. Thus, we define differences shown in equation (2) as the phenotypic plasticity trajectory of abundance for the $i$th strain. If there is variability in plasticity trajectory among $n$ strains, this implies the possible existence of QTLs that mediate phenotypic plasticity.

**Population structure and phylogenetic tree**. Based on a full set of SNP data, we use fastStructure software[106] to test the population structure of each GWAS population and use SNPhylo software[107] to build up the phylogenetic trees of S. aureus strains from each experiment (Fig. S1). We find eight and seven

subpopulations for the abiotic and biotic GWAS experiments, respectively (Fig. S2). A regression model is implemented to adjust the phenotypic data of microbial abundance at different time points, aimed at removing the confounding effect of population structure on our GWAS result. The adjusted phenotypes are used for the subsequent genetic analyses.

**Implementing coFunMap**. FunMap is a statistical model, proven to be powerful for mapping growth traits by integrating biologically meaningful growth equations[97,108–110]. More recently, Sang et al[98] have extended FunMap to coFunMap designed to map a composite trait mathematically constituted by different traits. We implement coFunMap to map plasticity QTLs. Consider a SNP $s$ with $J_s$ genotypes. Let $n_{js}$ denote the observation of genotype $j_s$ ($j_s = 1, \dots, J_s$). We formulate the likelihood of time-dependent differences of microbial abundance at this SNP as

$$L_s(\mathbf{z}) = \prod_{j_s=1}^{J_s} \prod_{i=1}^{n_{j_s}} f_s\left(\mathbf{z}_i; \boldsymbol{\mu}_{j_s}, \Sigma_s\right) \tag{3}$$

where $f_s(\cdot)$ is a $T$-dimensional normal distribution with mean vector $\boldsymbol{\mu}_{j_s}$ for genotype $j_s$ and residual covariance matrix $\Sigma$ at SNP $s$. Based on the definition, the mean vector can be expressed as

$$\boldsymbol{\mu}_{j_s} = (\mu_{j_s}(t_1), \dots, \mu_{j_s}(t_T)) \tag{4A}$$
$$= (\mu_{j_s}^x(t_1) - \mu_{j_s}^y(t_1), \dots, \mu_{j_s}^x(t_T) - \mu_{j_s}^y(t_T))$$

$$= (\mu_{j_s}^x(t_1), \dots, \mu_{j_s}^x(t_T)) - (\mu_{j_s}^y(t_1), \dots, \mu_{j_s}^y(t_T)) \tag{4B}$$

where $\mu_{j_s}(t)$ is the mean value of phenotypic plasticity for genotype $j_s$ of SNP $s$ at time point $t$ ($t = 1, \dots, T$) and $\mu_{j_s}^x(t)$ and $\mu_{j_s}^y(t)$ are the mean values of microbial abundance for genotype $j_s$ of SNP $s$ at time point $t$, expressed in environments X and Y, respectively. We consider that the time-dependent change of microbial abundance follows a S-shaped curve, as shown in equation (1). Thus, we model the genotypic means of microbial abundance at each genotype, specifically for each environment, as

$$\mu_{j_s}^x(t) = a_{j_s}^x \left[1 + b_{j_s}^x \cdot \exp\left(-r_{j_s}^x t\right)\right]^{-1} \tag{5A}$$

$$\mu_{j_s}^y(t) = a_{j_s}^y \left[1 + b_{j_s}^y \cdot \exp\left(-r_{j_s}^y t\right)\right]^{-1} \tag{5B}$$

where growth parameters ($a$, $b$, $r$) are genotype- and environment-specific, i.e., two sets of growth parameters for each genotype are used for curve fitting for two different environments.

The statistical power of coFunMap can also attribute to the covariance modeling[97,98,109,110]. The residual covariance matrix of phenotypic plasticity at SNP $s$ has the following structure:

$$\Sigma_s = \begin{pmatrix} \sigma_s^2(t_1) & \cdots & \sigma_s(t_1, t_T) \\ \vdots & \ddots & \vdots \\ \sigma_s(t_T, t_1) & \cdots & \sigma_s^2(t_T) \end{pmatrix} = \begin{pmatrix} \sigma_{sx}^2(t_1) & \cdots & \sigma_{sx}(t_1, t_T) \\ \vdots & \ddots & \vdots \\ \sigma_{sx}(t_1, t_T) & \cdots & \sigma_{sx}^2(t_T) \end{pmatrix} + \begin{pmatrix} \sigma_{sy}^2(t_1) & \cdots & \sigma_{sy}(t_1, t_T) \\ \vdots & \ddots & \vdots \\ \sigma_{sy}(t_1, t_T) & \cdots & \sigma_{sy}^2(t_T) \end{pmatrix} \tag{6}$$

where the time-dependent variance $\sigma_s^2(t)$ (and covariance $\sigma_s(t, t')$, $t, t' = 1, \dots, T$) of phenotypic plasticity are the sum of the time-variances $\sigma_{sx}^2(t)$ and $\sigma_{sy}^2(t)$ (and covariances $\sigma_{sx}(t, t')$ and $\sigma_{sy}(t, t')$) of the trait in two environments, respectively, under the assumption that two environments are independent of one another. Two environment-dependent covariance matrices in Eq. (6) contain longitudinal information and, thus, an autoregressive model, such as the first-order structured antedependence (SAD(1)) model[98], is used to fit the structure of the covariances, increasing the model's parsimony. The advantages of SAD(1) include the allowance of variances and covariances to change with time without need of additional parameters and the existence of the closed forms of matrix inverse and determinant favorable for efficient computation.

We implemented the simplex algorithm to obtain the maximum likelihood estimates (MLEs) of the model parameters constituting the likelihood (3). Using the MLEs of the parameters that model the mean vector (4 A), we can calculate the genetic standard deviation of a SNP $s$ for the phenotypic plasticity at any time point $t$, expressed as

$$g_s(t) = \sqrt{\frac{1}{n} \sum_{j_s=1}^{J_s} n_{j_s} \left(\mu_{j_s}(t)\right)^2 - \left(\frac{1}{n} \sum_{j_s=1}^{J_s} n_{j_s} \mu_{j_s}(t)\right)^2} \tag{7}$$

which describes the genetic effect of this SNP on the phenotypic plasticity of microbial growth. To test the statistical significance of this SNP, we formulate the following hypotheses:

$$H_0 : \boldsymbol{\mu}_{j_s} \equiv \boldsymbol{\mu} \tag{8}$$
$$H_1 : \text{Not all equalities in the } H_0 \text{ hold}$$

under each of which we calculate the likelihood values and use them to calculate the LR value as a test statistic. Permutation tests can be used to determine the genome-wide critical threshold. The procedures described above are used to identify key QTLs in traditional genetic mapping, but lacking a capacity to chart a complete picture of quantitative genetic architecture.

**Rationale of network reconstruction**. A better way of comprehending genetic control is to encapsulate all loci into an omnigenic network that shapes phenotypic variation. To do so, we need to introduce elements of multiple disciplines. We assume that all loci act and interact together to form a complex system, whose behavior can be interpreted through the lens of game theory. As a mathematical formulation of strategic interactions, game theory states that, to maximize its payoff, a rational player would decide an optimal strategy in response to the strategies of other players[111]. This decision process continues until the Nash equilibrium (i.e., strategically stable) is reached[112]. Combined with evolutionary theory, game theory is leveraged to evolutionary game theory, where evolutionarily stable strategies refine the Nash equilibrium without assuming players' rationality in interactions[44]. We integrate evolutionary game theory and predator-prey theory, an argument that describes the effects of predation on prey populations[45–49], to derive a generalized nLV equations for modeling "strategic" interactions among nonrational genes.

Let $\mathbf{g}_s = (g_s(t_1), \dots, g_s(t_T))$ denote the vector of marginal (net) genetic effects of SNP $s$ on phenotypic plasticity estimated by coFunMap. Based on evolutionary game theory, the net genetic effect of a SNP can be decomposed into its independent effect and dependent effect component[31]. This decomposition is mathematically specified by a system of nLV-based ordinary differential equations (ODEs), expressed as

$$\frac{dg_s(t)}{dt} = Q_s\left(g_s(t); \Theta_s\right) + \sum_{s'=1, s' \neq 1}^{p} Q_{s \leftarrow s'}\left(g_{s'}(t); \Theta_{s \leftarrow s'}\right) \tag{9}$$

where $dg_s(t)/dt$ is the derivative of the net genetic effect of SNP $s$ on phenotypic plasticity at time $t$, $Q_s(\cdot)$ is a time-varying function that characterizes the independent genetic effect of SNP $s$ that occurs when it is assumed to be in isolation, $Q_{s \leftarrow s'}(\cdot)$ is a time-varying function that characterizes the dependent genetic effect of SNP $s$ that arises from the influence of another SNP $s'$ on it, and $\Theta_s$ and $\Theta_{s \leftarrow s'}$ are a set of parameters that fit the independent and dependent functions, respectively. It can be seen that the independent genetic effect of a SNP depends on this SNP's intrinsic capacity (strategy), whereas its dependent genetic effect is determined by the genetic effect of the other SNP. Since genetic effects may not follow a parametric function, we implement a nonparametric smoothing approach, such as Legendre Orthogonal Polynomials (LOP)[113,114], to fit the independent and dependent functions.

Equation (9) represents a full model that allows a given SNP to be epistatically affected by all other $p - 1$ SNPs. However, such a full connection of epistasis in a cell is likely to be absent because this does not help the organism buffer against random perturbations and maintain its stability, as widely recognized in other network systems[68–71]. As such, the number of SNPs whose influences a given SNP can receive is limited, leading to the sparsity of genetic networks. A similar argument, called Dunbar's law, has been observed in animal social networks, where an individual can only stably handle a certain number of relationships with others due to a limited size of its neocortex—the part of the brain associated with cognition and language[115]. In statistics, to select a small set of SNPs that are the most significant for a given SNP, we incorporate regularization-based variable selection, such as LASSO and its variants[116–118]. This can be done by regressing the net genetic effect of a SNP on those of all other SNPs across time points. LASSO-related approaches can particularly handle variable selection when the number of SNPs is larger than the number of time points. In the case, where the relative number of SNPs over time points is so exceedingly large that LASSO may errantly perform, we may interpolate additional values on the fitting curve of genetic effects over time as a multiple regression model.

Through variable selection, we obtain a small set of the most significant genes for each gene. Let $d_s$ ($d_s \ll p$) denote the number of the most significant genes for gene $s$. By replacing $p$ by $d_s$ in the second term of equation (9), the full model is reduced to a sparse model, by which a sparsely-connected network can be reconstructed. We formulate a likelihood or a least-squares approach to solve the sparse ODEs by implementing the fourth-order Kunge-Kutta algorithm[50]. We code the estimates of independent genetic effects of different SNPs as nodes and the estimates of dependent genetic effects of gene pairs as edges into a graph. This graph represents a maximally informative networks filled of bDSW interactions[50].

**Detecting network communities through FunClu**. For a high-dimensional genetic network, sparsity is not only due to the limited number of links for a given SNP, but also to its division into delimited network communities. The existence of network communities is consistent with developmental modularity theory by which a complex system can be divided into different modules[72–74], a phenomenon

essential for the system to maintain its robustness and stability[68–71]. A module is defined as one whose components are more tightly with each other than those from other modules[119,120]. We implement a bottom-up approach for detecting network communities[50]. This approach is based on the assumption that SNPs within the same network communities have a more similar temporal pattern of genetic effects than those from other communities. In a GWAS of a trait measured at 500 time points, Verweig et al[84] found that SNPs are clustered based on the similarity of their genetic effects at all time points. Thus, we implement FunClu[77,78] to divide all $p$ SNPs into different modules according to the similarity of the temporal pattern of their genetic effects. This clustering allows us to reconstruct multiple interconnected smaller networks from a big network to overcome the curse of network dimensionality.

Let $\mathbf{g}_s = (g_s(t_1), \ldots, g_s(t_T))$ denote the genetic effect vector of SNP $s$ on phenotypic plasticity calculated by Eq. (7). Functional clustering is developed by formulating a mixture-based likelihood model, expressed as

$$L_c(\mathbf{g}) = \prod_{s=1}^{p} \sum_{l=1}^{L} \left[ \pi_l f_l(\mathbf{g}_s; \mathbf{u}_l, \Sigma_c) \right] \quad (10)$$

where $L$ is the number of modules, $\pi_l$ is a prior probability representing the proportion of module $l$, and $f_l(\cdot)$ is the $T$-dimensional multivariate normal distribution with mean vector $\mathbf{u}_l = (u_l(t_1), \ldots, u_l(t_T))$ for module $l$ and covariance matrix $\Sigma_c$. Since time-varying genetic effects may not follow an explicit equation, we implement LOP to model the structure of $\mathbf{u}_l$. We use SAD(1) to model the structure of $\Sigma_c$.

We implement a hybrid of the EM algorithm and simplex algorithm to solve the likelihood (10). An optimal number of mixture components (i.e., modules) is determined according to AIC. The module ($l$) to which a specific SNP $s$ belongs can be determined on the basis of the estimate of the posterior probability, expressed as

$$\Pi_{l|s} = \frac{\pi_l f_l(\mathbf{g}_s; \mathbf{u}_l, \Sigma_c)}{\sum_{l'}^{L} \pi_{l'} f_{l'}(\mathbf{g}_s; \mathbf{u}_{l'}, \Sigma_c)} \quad (10)$$

A SNP is assigned to a module if the posterior probability of this SNP ($\Pi_{l|s}$) within this module is larger than those in any other modules.

According to network theory, a module corresponds to a network community, which implies that functional clustering allows us to divide a big genetic network into distinct network communities. We first reconstruct a network that connects network communities based on the overall genetic effects of SNPs from the same modules. The node of this network, called the first-layer (top) network, is equivalent to the optimal number of modules by FunClu determined by information criteria, such as BIC. We then reconstruct genetic networks at the SNP level, called bottom networks, for each module, with the number of bottom networks being equal to the optimal number of modules. In current GWAS, it is not uncommon to have tens–thousands thousands of SNPs, in which case the number of SNPs within a module is still too big to reconstruct a meaningful SNP network. We further use FunClu, implemented with the LOP of a higher order, to classify such a module into its distinct submodules and reconstruct the second-layer network among these submodules. If needed, we classify a submodule into its sub-submodules and reconstruct the third-layer networks among these sub-submodules. This process continues until a small-size unit, allowing meaningful SNP networks to be reconstructed, is obtained. Taken together, we will reconstruct a multilayer, multiplex, and multiscale sparse network from any number of SNPs in a GWAS. While the top network is constituted by coarse-grained genetic interactions, the bottom network provides a detailed roadmap of how each SNP works together with every other SNP to determine phenotypic plasticity through fine-grained genetic interactions.

**Reporting summary**. Further information on research design is available in the Nature Research Reporting Summary linked to this article.

## Data availability
All data used in the manuscript are publicly available at https://github.com/CCBBeijing/PPMultilayerNetwork.

## Code availability
All code used in the manuscript is publicly available at https://github.com/CCBBeijing/PPMultilayerNetwork.

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

## Acknowledgements

The authors thank many members at the Center for Computational Biology for their contributions to collect data used in this manuscript. This work is supported by grant 31971398 from National Natural Science Foundation of China (X.H.).

## Author contributions

Computation, coding, and visualization: D.Y., A.D., J.W.; Experimentation: Y.J. and X.H.; Conceptualization, methodology, and writing: R.W. All authors contributed to manuscript editing and approved the manuscript.

## Competing interests

The authors declare no competing interests.
