## [Peer Review File · Nature Communications]

Peer Review Information

Manuscript title: Inferring multilayer interactome networks shaping phenotypic plasticity and evolution

Corresponding author name(s): Rongling Wu

Reviewer comments & decisions:

Reviewer comments, first version:

Reviewer #1 (Remarks to the Author: Overall significance):

The approach authors have taken to map the genetic effects, and relationships between them, on a bacterial phenotype is novel and interesting. The authors do a good job at citing previous methodological papers, but failed to cite relevant literature on the genetic basis of vancomycin non-susceptibility in *Staphylococcus aureus*, which is needed to contextualise and assess the biological plausibility of their findings.

See my full comments below:

In this study, Yang and colleagues study the genetic basis of differences in bacterial growth between two different experimental/environmental conditions, what they term 'phenotypic plasticity'. Specifically, they measure growth differences over time of *Staphylococcus aureus* strains exposed to vancomycin vs. not exposed in vitro. They next try to calculate the genetic effects of QTL, gene network modules and individual SNPs on growth differences. The current main limitations of the work are (1) the over-use of technical terms which makes it difficult to read and (2) the lack of biological interpretation and contextualisation of the findings. Specifically, given the a large scientific literature on the genetic basis of vancomycin non-susceptibility in *S. aureus*, a comparison of the discovered QTLs, and biological functions of gene modules, with known vancomycin-resistance-conferring genes is currently missing.

See individual comments below:

Methods and data availability:

Lines 794-800. Avoid repetition with introduction on the clinical relevance of *S. aureus* and vancomycin.

Details on bioinformatic pipelines to process bacterial genomic data, including SNP calling, is currently missing. Detailed information on software tools, versions and command lines used are needed.

Although the 'Data availability' statement indicates that 'All data and code used in the manuscript are publicly available', no link to a GitHub page or to other code repositories could be found in the manuscript, so it is not clear where the code can be found.

URL links to the repositories storing the genomic data should also be included. The 'serial number' [better named as 'accession number'] in Table S2 allowed me to find the genomic data on <https://bigd.big.ac.cn/bioproject/browse/PRJCA001035> and <https://bigd.big.ac.cn/bioproject/browse/PRJCA002084>. The authors should make sure the genomic data is also available on either the SRA NCBI or ENA EBI, as established software tools exist to retrieve genomic data from these widely used repositories.

Lines 812-827. The authors only made one MIC measurement available per strain in Table S2, while they reported more than one, specifically: (1) the initial MIC measurement of the parental strain (is this the one presented in Table S2?), (2) MICs after 60 days of in vitro treatment and (3) measurements of abundance [in what units?] at 14 time points "under two treatments, one containing 6 µg/mL vancomycin and the other being vancomycin-free (control)", that is, the source data of Figure 1A. All measurements must be made available to ensure reproducibility. The same applies for the growth measurements taken at 4 µg/mL and 2 µg/mL concentrations of vancomycin.

Results

Figure 1B. The term "phenotypic plasticity of microbial growth trajectory" described as the phenotype used in the GWAS may not be easy to interpret for the general reader. Please state in more plain terms what measurement was used as the outcome/phenotype variable in the GWAS, for example, "difference of microbial abundance between control and stress conditions", or, as stated in the discussion "difference of growth trajectory" between conditions. Also, coFunMap, the method used to map QTLs, is not described in the Methods, please add citation to the method/tool and details of parameters used. It is also important that QTLs are annotated on the Manhattan plot with their gene name and nucleotide or amino acid change, so that specialised readers can recognise genes.

Lines 114-125. The authors note that strains differ notably in the way they grow under vancomycin stress, compared to how they grow in vancomycin-free medium. Presumably, strains displaying a S-shape

growth under vancomycin treatment may be those that are resistant to vancomycin (i.e. exhibit an elevated parental MIC to vancomycin). Can the authors check and discuss whether this is the case? In other words, do the vancomycin MIC of parental strains explain difference in growth between conditions?

Lines to 127 to 138. There is quite a large scientific literature on the genetic basis of vancomycin non-susceptibility in *S. aureus*. Even though the phenotype measured here (growth differences in vancomycin vs. vancomycin-free medium) is different from that commonly measured in other studies (vancomycin susceptibility as measured by minimum inhibitory concentrations at a single time point), it would be relevant to compare if there is any overlap in the genes described in the literature contributing to vancomycin resistance, and the QTL identified here. Unfortunately, finding that “candidate genes that determine a variety of cellular and molecular processes” is not enough to suggest “the biological relevance of functional mapping.”

Line 153. “Detection of network communities” I am more familiar with the term “module” than “community” to refer to groups of genes involved in the same biological function (e.g. metabolic modules). The title of this section could be re-named as “Detection of network modules”.

Lines 153 – 178. In order to assess the biological relevance of these 14 modules, and sub-modules described later on, the authors must annotate the biological function of genes within each module and sub-module (such as KEGG terms, metabolic function, etc.). Also, further details on each module like their size (number of genes) and enriched biological function is needed. This could be presented as a supplementary table.

Line 176. Check for typo in sentence: “communities, especially M7, M12, and M12, display a similar pattern of time-varying genetic effects.” as M12 is repeated twice.

Lines 212-259. Biological interpretation of findings is lacking in these paragraphs. The gene name, function and amino acid change of SNPs, QTLs and sub-modules presented here would allow reviewers and readers to assess the biological plausibility of the relationships (inhibition and promotion) described between them.

Reviewer #1 (Remarks to the Author: Impact):

I think major revisions are still needed to judge what Nature journal this paper would be most suitable for.

Reviewer #1 (Remarks to the Author: Strength of the claims):

See full comments above.

Reviewer #1 (Remarks to the Author: Reproducibility):

Regarding reproducibility, the authors do not provided the code and the source data needed to reproduce their findings, see specific queries in my full comments above.

Reviewer #2 (Remarks to the Author: Overall significance):

Summary: This manuscript proposes a novel method of disentangling gene regulatory networks from time series genetic mapping (QTL/GWAS) data. It is applied and discussed with respect to understanding the developmental and genetic bases of phenotypic plasticity and is presented via analysis of bacterial resistance to vancomycin. The method proposed seems to better capture gene interactions than current approaches to analyzing mapping data.

I was very excited reading this paper and immediately began thinking about how this method could impact my own work in the future. I cannot really speak to the validity or appropriateness of the computations and calculations underlying the method, but the big picture and example presented in the manuscript suggest to me that this will be a powerful tool for disentangling the bases of complex phenotypes. I especially can see it influencing the way evolutionary biologists approach developmental questions.

Reviewer #2 (Remarks to the Author: Impact):

I think this novel method/framework will influence how the genetic bases of complex traits is investigated if the authors can make their methods user-friendly and accessible.

Reviewer #2 (Remarks to the Author: Strength of the claims):

I am convinced that this work could be generally useful. However, I do have three major comments and various minor comments that I think would improve the manuscript, make it more transparent, and more user-friendly (especially for this approach to be widely adopted).

1) First, it is not clear to me why the authors focus so heavily on phenotypic plasticity. It seems to me that this approach should be applicable to any trait for which there is time-series data. Relatedly, it wasn't even clear to me how the authors calculated plasticity in this study (i.e., how were specific values for specific timepoints and strains determined?) While I certainly think a plasticity context is interesting (I am biased in this regard), I think the manuscript could be strengthened if the authors broaden their

scope to simply encompass development of any trait and, perhaps, briefly discuss plasticity. Maybe I missed something special about this approach that makes it plasticity-centric.

2) Second, the authors should definitely include a table that describes the necessary inputs for this approach. It took me a while to realize/comprehend that time course data was needed. Such data is not always available—especially for mapping—in many study systems. Therefore, I think a paragraph or two needs to be added to the discussion that address the potential limitations of this method. For example, it may not work in the case of polyphenisms where bimodal or binary phenotypic differences are not observed until adulthood (i.e., in cases where longitudinal data are especially hard to acquire and that have limited windows for variation to be detected).

3) Third, the hands-on side of the manuscript should be bolstered if the authors desire to have this tool used. Certainly, the description of the approach and its application to the empirical data they use is great, but I could not envision how to actually use this method for my own work. I know that data and code would presumably be available upon acceptance for publication, but the authors should consider how they plan (if at all) to make this method readily implementable by fellow researchers. Doing so would greatly strengthen the impact of this paper and bring about the transformative change the authors describe. As it presently stands, this approach brings together various ways of thinking into a novel conceptual approach, but not a novel practical approach. I sincerely hope it becomes the latter as well.

My minor comments are below:

4) At various places in the manuscript the font color changes unexpectedly from black to a more grayish color. Presumably, this is the result of editing or copying from other drafts/communications.

5) L129-130 Please related this description of developmental plasticity more explicitly to the preceding sentence. Related to part of my point 1 above, plasticity was simply the difference between control and stress and then this was taken at various timepoints in development, correct?

6) L131 Please define or describe what composite functional mapping is and if it is unique to your framework

7) L172 Change “A total of” to “The”

8) L173 and ‘than’ after the word ‘rather;

9) L176 M12 is listed twice

10) L189 How were these 45 links identified as actually occurring

11) L191 Is it SNPs or modules at this point?

12) L196 change ‘that trigger’ to ‘have’

13) L197-198 This is a cool level of detail to be able to get to!

14) L223-224 This predictive and hypothesis-driving potential could be very useful

15) L242 ‘receive more incoming links that send outgoing links’

- 16) L256 remove 'we actually have'
- 17) L266 remove 'how'
- 18) L276 'leadership'
- 19) L411 italicize *S. aureus*
- 20) L416 italicize *S. aureus*
- 21) L449 remove 'but'
- 22) L459-460 Remove 'published...University' and just include the citations
- 23) Methods I know the journal dictates manuscript formatting, but I think this manuscript could greatly benefit from having the methods before the results or co-mingled. This is especially important because of the tool-like nature of this manuscript. Making it easier to see how what was done relates to what was found would be very useful for this manuscript.
- 24) Figure 1A rightmost panel: I am not sure what to make of this figure. Is it meant to show how plasticity was calculated? If so, I am still confused and perhaps more description is warranted.

Reviewer #2 (Remarks to the Author: Reproducibility):

As indicated elsewhere in my review, I think the methods could be made more user-friendly and hands-on so that those not deeply involved in the concepts/theory could implement the approach.

Reviewer #3 (Remarks to the Author: Overall significance):

Review of Inferring multilayer interactome networks shaping phenotypic plasticity and evolution by Yang, Jin, He, and Wu

Main comments

Yang et al. attempt to deduce networks of interactions between loci underpinning a plastic trait. Their goal is laudable and ambitious, and the experimental aspect of their work is fascinating on its own. However, this manuscript, in its current form, suffers from a number of shortcomings and, in my view, requires extensive revisions before it reaches a publishable unit. Below, I highlight some of the major shortcomings of the manuscript in its current form.

Reviewer #3 (Remarks to the Author: Strength of the claims):

First, the premise of the experiment is not well-justified. Specifically, how does one disentangle the plastic response from the genetic response, given the fact that GWAS, by definition, is linking phenotypic

variance directly to the genetic variation? In the absence of such justifications, statements such as “... tremendous inter-strain variation in the pattern ... implies the existence of genes that govern phenotypic plasticity.” (lines 123-134) are not warranted.

Second, the computational model to infer the so-called omnigenic interactome networks, which is the crux of the manuscript, is explained in vague terms. In the introduction section (lines 81 - 94), evolutionary game theory and network theory are invoked, without even the most cursory explanation as to why these theories were utilized to infer interactions from QTLs deduced from a GWAS study. The method section does very little to illustrate how the model works. Statements such as “according to network theory” (line 775) are extremely confusing as they imply that the network theory is something akin to the second law of thermodynamic, while in fact, the sentence is as much useful as a sentence that starts with “according to the theory of evolution ...”. More precise language and more extensive explanations of the model are required.

Third, the section title “Detection of network communities” (lines 153 - 177), while being crucial for understanding the manuscript, lacks the necessary explanations and elaboration. The developmental modularity theory is introduced without any explanations. (as side note: I presume that BIC (line 166) stands for Bayesian information criterion, but acronyms should always be introduced properly.)

Finally, it is not at all clear how measuring genetic effects over 50 hours is a justifiable method to deduce the gene regulatory network that underlies a phenotype when the population in question has undergone roughly 100 generations already (assuming generation time of 30 min). How does adaptive evolution in the presence of antibiotics affect the topology of the interactome networks inferred from the QTLs? In the absence of such justifications, the biological nature and significance of the inferred interactions, such as the supposed “antagonistic” relationship between QTL2728755 and NP2748895, remains elusive. The extensive discussion about the nature of interactions between QTLs and SNPs, as promoters or suppressors, is similarly affected.

Aside from the aforementioned comments, given the complexity of the model proposed in this manuscript, it's a pity that its poor structure prevents an adequate comprehension and a fair assessment of its central claim.

Minor points

Line 70: Suggest citing some of the famous papers on the definition of epistasis, including Patrick Philips' The language of gene interaction (GENETICS July 1, 1998 vol. 149 no. 3 1167-1171).

Line 71: “statistical approaches used in the literature cannot reveal the causality of epistasis and the sign causality” => causality not causality.

“..., consistently with a larger amount of growth ...” (Line 116) is a slightly misleading characterization of the growth curves in Figure 1, as most strains reach similar levels of abundance under stress compared

to the control.

Ref 43 should be Maynard Smith, J. and not Smith, J. M.

In “ ... implying their neglected role ...” (144) => probably “negligible” and not “neglected”.

Line 455: It is either “a wealth of literature” or “a body of literature”, but not both.

Line 682: “differences in equation (1)” => “difference equation (1)”

Line 752: “a hybrid of the EM algorithm” => a hybrid of the expectation–maximization (EM) algorithm”

Reviewer #4 (Remarks to the Author: Overall significance):

In this paper, the authors analyse the genetic basis of *S. aureus* resistance to vancomycin by using a statistical approach based on the omnigenic model. They measured growth curves of 99 strains with and without this stress, and used a genome-wide association method for time series data to find significant core SNPs controlling this response. The authors then applied evolutionary game theory to construct an interacting network of core SNPs which can explain the measured response curves. Whereas a ‘typical’ genetic model of this phenotype would consist solely of the five independently significant QTLs discovered in the first phase; in this model, insignificant SNPs typically have large individual effects on the phenotype, but most of these cancel out due to regulatory effects from other core SNPs in their network. These interactions can be measured and, in some cases, interpreted. The authors imply that many of the regulatory SNPs would be common actors in a range of phenotypes, though this is not tested.

The omnigenic theory of trait inheritance is an interesting one, worthy of further study in species other than humans. I believe that the statistical approaches and experiments presented here are broadly appropriate to address this question in a limited way. However, with current manuscript I had the following issues with its presentation:

- Broad, sweeping claims are made about the approach being a ‘paradigm-shifting’ framework. However, only a single phenotype, in a single species, in a specific strain collection was tested. A key feature of the omnigenic model is that the regulatory SNPs regulate many phenotypes, but as the authors only test one phenotype they cannot identify whether the links in their regulatory networks are shared.
- The omnigenic model is essentially taken as a prior for the analysis here. All SNPs are included from the bottom-up, as a feature. But what about other models? Does this model the data better than say a model with the QTLs? Or a polygenic model such as GCTA and LDAK? Without these comparisons, I do not think that the authors can claim that their model and approach is superior in terms of predictive power or mechanistic understanding of these traits.
- There is not enough clarity, particularly in the introduction and conclusion, on what it actually is that the authors are testing. There are a lot of buzzwords, but from my reading the main thing this

manuscript does show us one way to construct an omnigenic model for a trait, and compare it with some of the QTLs found in a GWAS. I would encourage the authors to spend more time introducing monogenic, polygenic, omnigenic models, and missing heritability. Give the biological/genetic interpretations of each, and compare and contrast them. Additionally, note the difference between phenotype prediction and understanding the biological basis of phenotypes – two related but different goals of these models.

- There is a long part of the paper and the figures showing and describing links in networks. Without either strong evidence that this model is superior, or biological interpretation (SNPs in this section are simply numbered), this is both speculative and offers poor explanatory power.
- The section on ‘model validation’ was not particularly convincing. It mostly seemed to repeat the same experiment, as the essentially the same phenotype and analysis were redone. This felt like a bit of a missed opportunity: if the authors had made mutants which can directly measure effects of SNPs, they could have tested regulatory predictions from the constructed networks, and compared these with a model of individual QTLs. At least, comparing predictive power of the network approach versus the QTL approach in a new dataset would have been more valuable.

Reviewer #4 (Remarks to the Author: Impact):

I do think that this paper could be an interesting addition to the microbial statistical genetics literature. As far as I am aware, no one has previously explored the omnigenic model in bacteria before. But to do so effectively, the authors need to be less bold in their claims, clearly compare the genetic models they are testing, and be refocus to target a comparison between omnigenic and polygenic models specifically.

Overall, I can see the rationale for this model, but is based on asserting that all SNPs have an effect: can we connect these in a way to get the pattern in this (single) GWAS? I don't see evidence here that this model refutes just the significant QTLs, or a standard polygenic model, predicting the data equally well.

For a more impactful paper which would inspire others to use this model, I would expect to see more targeted experiments, using new strains or mutants, which tested these models predictive and explanatory power. I would also expect to see more reusable code, which include some continuous integration testing.

Reviewer #4 (Remarks to the Author: Strength of the claims):

Required:

1) The effects of population structure are dealt with only in passing, but are a major issue for many GWAS studies. I agree that the QQ plot and Manhattan plot look broadly correct, but would need to also

see:

- a. A phylogenetic tree of the isolates, ideally with some summary of the phenotype mapped on as a trait.
- b. More information of how the eight subpopulations were identified and removed.
- c. A comparison of a standard linear mixed model versus a quantitative summary statistic of the trait (for example the growth rate and/or carrying capacity)
 - 2) More biological inference of the *S. aureus* results integrated into the main text, moving table S1 into the main results, and describing SNPs in terms of known function rather than just position. In particular, a biological interpretation of the modules of SNPs would make the model more convincing.
 - 3) Make a clear comparison between the omnigenic model of trait heritability, and a mono/polygenic model. Show which has better explanatory power. I would note that more sub-significant SNPs affecting a phenotype is well established and modelled by e.g. the GCTA model. Implicitly, the GCTA model assumes that all SNPs have small effects. So the difference with the omnigenic model is that many have large effects, but they cancel out through interactions. But, which is the model better supported by the data here?
 - 4) In missing heritability, calculating h^2 , and h^2 from the QTLs would be a really helpful way to compare these models. Adding the significant SNPs from the module approach to a h^2 calculation would also be informative.
 - 5) Likewise, using a method such as SpyrPick or just simple associations of interaction terms to find epistasis via 'conventional' means would be useful to compare to interactions in the networks (even if the result is that this is underpowered).
 - 6) The cancelling out of strong effects through interactions in the omnigenic model is necessary to reconcile individual effects at the lower level with no effect overall (the QTLs in the GWAS). What would the alternative look like? Is there a way to not have an omnigenic model and still get the GWAS effects?

Optional

- 1) The authors only examine SNPs in the core genome, but the accessory genome is known to have an important role in adaptive response in bacteria (e.g. <http://dx.doi.org/10.1038/s41559-017-0337-x>, and many others). Could the accessory genome variation be included in an omnigenic model?
- 2) If the same regulatory loci and interactions were found to hold for other phenotypes in the species, the model and approach here would be more convincing. A multi-phenotype study, perhaps RNA-seq, would be most powerful of all. There are some public datasets available with this kind of data.
- 3) Validation of the model by creating mutants based on the networks speculated above, to test whether these SNPs do affect regulation would add a lot. For example, can both the network approach, and QTL approach be used to predict phenotype response in a recombinant mutant. Where these predictions are divergent, then testing which phenotype the mutant has would be a powerful confirmation of that model.

Additional minor comments:

- 1) In the introduction: omnigenic theory is describe as all traits being controlled by all genes. My understanding is that there are common regulatory genes which are involved in many phenotypes, not that every gene is functional. How could the authors reconcile their assertion that all genes are involved with the frequent loss of genes in bacteria?
- 2) Network is used many times with a specific definition in the introduction.
- 3) 'millions of millions of gene pairs' – in a typical bacterial population there would be around 10^7 genes. Be more specific if possible. Also, this is consistently stated to be computationally intractable. Networks with this many edges are possible to analysed however.
- 4) Inferring this many interactions from a few samples is still likely to be underpowered, even with improved analysis (even though the time series does add power). How confident are the authors that their network is the only correct one?
- 5) 'Statistical approaches cannot infer the casuality of epistasis'. Is this specific to epistasis? Approaches such as Mendelian randomization, or that of Pearl et al have attempted this I believe. Also a typo in 'causality'.
- 6) 'design an efficient and effective gene editing program'. In bacteria? More broadly?
- 7) A deficiency of linear additive models is stated as missing random effects, but linear mixed models, the most commonly used models in GWAS and breeding, do include random effect terms.
- 8) Is bDSW a standard acronym? I found its use confusing, and would suggest that simply 'edges' or 'interactions' would be easier to follow.
- 9) I noted that the genetic effect vs time curves mostly show effect during the growth phase (rather than stationary). Is this because this is where phenotype difference/derivative with respect to time is greatest, and there is greatest power?
- 10) 'find that their effect curves are quite flat, implying their neglected role in shaping phenotypic plasticity.' How does this implication follow?
- 11) I appreciate that minimizing BIC is a good quantitative way to choose the number of modules, but it might be informative to use a lower number, and see if any obvious biological pattern is apparent (as is, many of the module effects look very similar).
- 12) The beginning of the section 'multilayer genetic networks' could more clearly and succinctly be described as 'we recursively subdivide communities'.
- 13) On sparsity: is finding a sparse network a necessary property the method of subdivision, or a finding when applied to this dataset?
- 14) 'Most links are directional rather than reciprocal' – invoking interactions between populations of animals is unlikely to be universal, does it really apply to SNPs and antimicrobial resistance?
- 15) The discussion restates a lot of the results, and could be trimmed. Some of the biological interpretation should be in the results.
- 16) 'Our model integrates the theories of two recent perspective articles published in Cell by J. Pritchard and M. Snyder both at Stanford University'. The citations will suffice.

Reviewer #4 (Remarks to the Author: Reproducibility):

1) The code URL isn't listed in the manuscript, but I found it (with a typo) in the reporting summary. In such a manuscript, the code needs to be more prominent, and in a state where others can use it. At the moment it is fairly unstructured, has local links to files, and could certainly not be reused or replicated by others.

2) Analysis in this paper was written from scratch, no existing software was used. How can we be sure that no errors occurred in this process? The final model is complex. Some testing of the code itself is needed.

Author rebuttal, first version:

Reviewer #1 (Remarks to the Author: Overall significance)

The approach authors have taken to map the genetic effects, and relationships between them, on a bacterial phenotype is novel and interesting. The authors do a good job at citing previous methodological papers, but failed to cite relevant literature on the genetic basis of vancomycin non-susceptibility in *Staphylococcus aureus*, which is needed to contextualise and assess the biological plausibility of their findings.

Our response: Thank you for your constructive comments. Citing more relevant literature on the genetic basis of vancomycin non-susceptibility in *Staphylococcus aureus* is a reasonable point. We have adopted this suggestion to cite and discuss such literature.

See my full comments below:

In this study, Yang and colleagues study the genetic basis of differences in bacterial growth between two different experimental/environmental conditions, what they term 'phenotypic plasticity'. Specifically, they measure growth differences over time of *Staphylococcus aureus* strains exposed to vancomycin vs. not exposed in vitro. They next try to calculate the genetic effects of QTL, gene network modules and individual SNPs on growth differences. The current main limitations of the work are (1) the over-use of technical terms which makes it difficult to read and (2) the lack of biological interpretation and contextualisation of the findings.

Specifically, given the a large scientific literature on the genetic basis of vancomycin non-susceptibility in *S. aureus*, a comparison of the discovered QTLs, and biological functions of gene modules, with known vancomycin-resistance-conferring genes is currently missing. See

individual comments below:

Our response: Our manuscript is defined as a Methods paper, which presents a new analytical strategy for dissecting phenotypic plasticity by combining several disciplines. We agree with you that to understand this paper, we need knowledge from other disciplines. For example, “network community” is a very common concept in network science, but may be new to general biologists. Our work is focused on the reconstruction of genetic networks and, therefore, it is natural to use and introduce this concept into the paper. Biological relevance of our discoveries has been analyzed by performing extensive KEGG gene enrichment analyses and pathways analysis.

We have cited several studies of the genetic analysis of vancomycin resistance in *S. aureus*. It should be pointed out that the gene identification of this phenomenon is mostly based on mutagenesis or genome sequencing, having little literature on genetic dissection using GWAS.

Methods and data availability:

Lines 794-800. Avoid repetition with introduction on the clinical relevance of *S. aureus* and vancomycin.

Our response: We have deleted some repetition in this part.

Details on bioinformatic pipelines to process bacterial genomic data, including SNP calling, is currently missing. Detailed information on software tools, versions and command lines used are needed.

Our response: We have provided all the information about bioinformatic pipelines in this revised version. Genomic DNA of all strains were extracted by TIANamp Bacteria DNA Kit (TIANGEN, Beijing, China) according to the manufacturer’s protocol. Whole-genome sequencing was performed using Illumina HiSeq 4000 instrument (Illumina Inc., San Diego, CA, United States) at Allwegene (Beijing, China). To obtain initial alignment results, sequencing data was compared with the reference genome (*S. aureus* subsp. *aureus* NCTC 8325) using BWA mapper v0.7.8 (Li, 2013). SAMtools software package v0.1.18 (Li et al., 2009) was used to sort the alignment results and obtain SNPs. More details have been added in the Methods section.

References

Li, H. (2013). Aligning sequence reads, clone sequences and assembly contigs with BWA-MEM. arXiv [Preprint]. arXiv:1303.3997

Li, H., Handsaker, B., Wysoker, A., Fennell, T., Ruan, J., Homer, N., et al. (2009). The sequence alignment/map format and SAM tools. *Bioinformatics* 25, 2078–2079.

Although the ‘Data availability’ statement indicates that ‘All data and code used in the manuscript are publicly available’, no link to a GitHub page or to other code repositories could be found in the manuscript, so it is not clear where the code can be found.

Our response: All code and data are uploaded to a public site at <https://github.com/CCBBEijing/PPMultilayerNetwork>.

URL links to the repositories storing the genomic data should also be included. The ‘serial number’ [better named as ‘accession number’] in Table S2 allowed me to find the genomic data on <https://bigd.big.ac.cn/bioproject/browse/PRJCA001035> and <https://bigd.big.ac.cn/bioproject/browse/PRJCA002084>. The authors should make sure the genomic data is also available on either the SRA NCBI or ENA EBI, as established software tools exist to retrieve genomic data from these widely used repositories.

Our response: We have changed “serial number” as “accession number.” Also, all the genomic data have been deposited in NCBI repository at <https://www.ncbi.nlm.nih.gov/Traces/study/?acc=PRJNA722566>. As suggested, we have submitted the genomic data to SRA NCBI. The Serial numbers were included in Table S1.

Lines 812-827. The authors only made one MIC measurement available per strain in Table S2, while they reported more than one, specifically: (1) the initial MIC measurement of the parental strain (is this the one presented in Table S2?), (2) MICs after 60 days of in vitro treatment and (3) measurements of abundance [in what units?] at 14 time points “under two treatments, one containing 6 µg/mL vancomycin and the other being vancomycin-free (control)”, that is, the source data of Figure 1A. All measurements must be made available to ensure reproducibility. The same applies for the growth measurements taken at 4µg/mL and 2µg/mL concentrations of vancomycin.

Our response: All *S. aureus* strains (numbered as S1–S9, S11–S36, S40–S45, S47–S85, S87–S105) were obtained from parental strains by 60-day in vitro vancomycin treatment and MICs were listed in Table S1. All parental strains were VSSA (MIC=1-1.5) and numbered as S_p1–S_p9, S_p11–S_p36, S_p40–S_p45, S_p47–S_p85, S_p87–S_p105 (Table S1). The MIC of parental strains were included in the revised Table S1. The OD₆₀₀ was measured at different time points to represent growth abundance. To address the reviewer’s concern, more details have been added on page 31.

Results

Figure 1B. The term “phenotypic plasticity of microbial growth trajectory” described as the phenotype used in the GWAS may not be easy to interpret for the general reader. Please state in more plain terms what measurement was used as the outcome/phenotype variable in the GWAS,

for example, “difference of microbial abundance between control and stress conditions”, or, as stated in the discussion “difference of growth trajectory” between conditions. Also, coFunMap, the method used to map QTLs, is not described in the Methods, please add citation to the method/tool and details of parameters used. It is also important that QTLs are annotated on the Manhattan plot with their gene name and nucleotide or amino acid change, so that specialised readers can recognise genes.

Our response: We have provided the definition of phenotypic plasticity which is the difference of microbial abundance for the same strain expressed in control and stress. Also, we have given a detail on coFunMap and its citation.

In the Manhattan plot, the names of genes associated with QTLs are indicated.

Lines 114-125. The authors note that strains differ notably in the way they grow under vancomycin stress, compared to how they grow in vancomycin-free medium. Presumably, strains displaying a S-shape growth under vancomycin treatment may be those that are resistant to vancomycin (i.e. exhibit an elevated parental MIC to vancomycin). Can the authors check and discuss whether this is the case? In other words, do the vancomycin MIC of parental strains explain difference in growth between conditions?

Our response: Those that grow in an S shape in stress imply their high vancomycin resistance, marginally associated with an elevated parental MIC ($P = 0.12$) (Table S1).

Lines 127 to 138. There is quite a large scientific literature on the genetic basis of vancomycin non-susceptibility in *S. aureus*. Even though the phenotype measured here (growth differences in vancomycin vs. vancomycin-free medium) is different from that commonly measured in other studies (vancomycin susceptibility as measured by minimum inhibitory concentrations at a single time point), it would be relevant to compare if there is any overlap in the genes described in the literature contributing to vancomycin resistance, and the QTL identified here. Unfortunately, finding that “candidate genes that determine a variety of cellular and molecular processes” is not enough to suggest “the biological relevance of functional mapping.”

Our response: We have cited the literature on the genetic analysis of vancomycin non-susceptibility in *S. aureus*. This type of studies is mostly based on in vitro mutagenesis and sequencing-based approaches, with a little based on GWAS. We have provided more detailed information about gene enrichment analysis of the QTLs detected and their comparison with the genes described in the literature. For example, two annotated QTLs Q550312 and Q550323 reside in the region of gene *SAOUHSC_00544 (SdrC)* that codes fibrinogen-binding protein. In a genomic analysis of highly vancomycin-resistant *S. aureus* strains developed by in vitro mutagenesis, gene *SdrC* was identified to play an important role in vancomycin resistance (Ishii et al. 2015; Ameri and Cooper 2019). QTL Q25261 is located at gene *SAOUHSC_00020 (walR)*

or *yycG*) encoding cell envelope biogenesis (Johnston et al. 2016). *walRK* phosphorylation activates the expression of YycH and YycI to control cell wall metabolism and biofilm formation in *S. aureus* (Dubrac et al. 2007; Gajdiss et al. 2020).

Line 153. “Detection of network communities” I am more familiar with the term “module” than “community” to refer to groups of genes involved in the same biological function (e.g. metabolic modules). The title of this section could be re-named as “Detection of network modules”.

Our response: “Network communities” is a very common concept in network science. Our work is focused on the reconstruction of genetic networks and, therefore, it is natural to introduce this concept into the paper. “Network communities” and “network modules” are actually interchangeable, although the former is more popular to network researchers. Considering this balance, we incline to use “network communities.”

Lines 153 – 178. In order to assess the biological relevance of these 14 modules, and sub-modules described later on, the authors must annotate the biological function of genes within each module and sub-module (such as KEGG terms, metabolic function, etc.). Also, further details on each module like their size (number of genes) and enriched biological function is needed. This could be presented as a supplementary table.

Our response: We adopted this comment. We provided the number of genes within each module (Fig. 3A) and enriched biological function of each module (Fig. 3B).

Line 176. Check for typo in sentence: “communities, especially M7, M12, and M12, display a similar pattern of time-varying genetic effects.” as M12 is repeated twice.

Our response: Thank you for pointing out a typo here. One of M12 is M10, which has been corrected.

Lines 212-259. Biological interpretation of findings is lacking in these paragraphs. The gene name, function and amino acid change of SNPs, QTLs and sub-modules presented here would allow reviewers and readers to assess the biological plausibility of the relationships (inhibition and promotion) described between them.

Our response: This part is one of the most important results for this paper. We have performed extensive gene enrichment analysis for modules, QTLs and SNPs and found several biologically interesting results. For example, our model shows that QTLs Q550323 and Q183488 have large independent genetic effects. It is found that these QTLs are located at candidate genes (*SdrC* for Q550323 and *oppA* for Q183488) that are directly involved in antibiotic resistance, which thereby explains the reason why they have remarkable independent effects. Also, several SNP-SNP interactions detected by our model can be well annotated to protein-protein interactions

through KEGG analysis. In other words, a pair of proteins encoded by linked genes in SNP networks reconstructed by our model are correspondingly linked in protein networks. These lines of evidence can well explain the validity of our model.

As you will see, we have almost rewritten this part to clearly explain the results from gene annotation analysis.

Reviewer #1 (Remarks to the Author: Impact):

I think major revisions are still needed to judge what Nature journal this paper would be most suitable for.

Our response: We hope that we have satisfactorily answered your comments.

Reviewer #1 (Remarks to the Author: Strength of the claims):

See full comments above.

Reviewer #1 (Remarks to the Author: Reproducibility):

Regarding reproducibility, the authors do not provided the code and the source data needed to reproduce their findings, see specific queries in my full comments above.

Our response: We have provided all the code and data at a public site at <https://github.com/CCBBeijing/PPMultilayerNetwork>.

Reviewer #2 (Remarks to the Author: Overall significance):

Summary: This manuscript proposes a novel method of disentangling gene regulatory networks from time series genetic mapping (QTL/GWAS) data. It is applied and discussed with respect to understanding the developmental and genetic bases of phenotypic plasticity and is presented via analysis of bacterial resistance to vancomycin. The method proposed seems to better capture gene interactions than current approaches to analyzing mapping data.

I was very excited reading this paper and immediately began thinking about how this method could impact my own work in the future. I cannot really speak to the validity or appropriateness of the computations and calculations underlying the method, but the big picture and example presented in the manuscript suggest to me that this will be a powerful tool for disentangling the bases of complex phenotypes. I especially can see it influencing the way evolutionary biologists approach developmental questions.

Our response: Thank you for your constructive and encouraging comments. We have attempted to validate our model by extensive gene enrichment analysis. For example, our model shows that QTLs Q550323 and Q183488 have large independent genetic effects. It is found that these QTLs are located at candidate genes (*SdrC* for Q550323 and *oppA* for Q183488) that are directly involved in antibiotic resistance, which thereby explains the reason why they have remarkable independent effects. Also, several SNP-SNP interactions detected by our model can be well annotated to protein-protein interactions through KEGG analysis. In other words, a pair of proteins encoded by linked genes in SNP networks reconstructed by our model are correspondingly linked in protein networks. These lines of evidence can well explain the validity of our model.

We have conducted an additional independent GWAS experiment, with results that can be well explained from a biological perspective. This reasonably supports the transparency and reproducibility of our model.

Reviewer #2 (Remarks to the Author: Impact):

I think this novel method/framework will influence how the genetic bases of complex traits is investigated if the authors can make their methods user-friendly and accessible.

Our response: This is a great point. We have packed our method into an R script and uploaded at a public site <https://github.com/CCBBeijing/PPMultilayerNetwork>, from which researchers worldwide can freely download and use our method. We will make every effort (including presentations to the conferences, seminars, and workshops) to disseminate our method and publicize its use to a range of biological, evolutionary, and medical fields.

Reviewer #2 (Remarks to the Author: Strength of the claims):

I am convinced that this work could be generally useful. However, I do have three major comments and various minor comments that I think would improve the manuscript, make it more transparent, and more user-friendly (especially for this approach to be widely adopted).

1) First, it is not clear to me why the authors focus so heavily on phenotypic plasticity. It seems to me that this approach should be applicable to any trait for which there is time-series data. Relatedly, it wasn't even clear to me how the authors calculated plasticity in this study (i.e., how were specific values for specific timepoints and strains determined?) While I certainly think a plasticity context is interesting (I am biased in this regard), I think the manuscript could be strengthened if the authors broaden their scope to simply encompass development of any trait and, perhaps, briefly discuss plasticity. Maybe I missed something special about this approach that makes it plasticity-centric.

Our response: Our method can be used to analyze any complex traits with longitudinal measurements. The motivations to focus on phenotypic plasticity include (i) phenotypic plasticity is an important topic by itself and there is a huge body of literature on this, (ii) we have just conducted GWAS experiments with multiple environments so that the data from these experiments can be directly analyzed, and (iii) phenotypic plasticity is more complex than a trait by definition. Because phenotypic plasticity is a complex trait, the application of our model to other traits is straightforward.

2) Second, the authors should definitely include a table that describes the necessary inputs for this approach. It took me a while to realize/comprehend that time course data was needed. Such data is not always available—especially for mapping—in many study systems. Therefore, I think a paragraph or two needs to be added to the discussion that address the potential limitations of this method. For example, it may not work in the case of polyphenisms where bimodal or binary phenotypic differences are not observed until adulthood (i.e., in cases where longitudinal data are especially hard to acquire and that have limited windows for variation to be detected).

Our response: To demonstrate the procedure of deriving our approach from GWAS data, we draw a flowchart (Fig. 1), showing what types of data and knowledge are required to reconstruct multilayer networks.

Our approach needs availability of longitudinal data. We admit that longitudinal data are not more common than static data, but they have been increasingly available with the advent of high-throughput automatic or semi-automatic phenotyping techniques. As a first step of this work, we derive our model based on more informative longitudinal data. Next, we implement more sophisticated statistical and mathematical theories to derive similar networks based on static data. We have discussed the limitations of our approach in the Discussion section.

3) Third, the hands-on side of the manuscript should be bolstered if the authors desire to have this tool used. Certainly, the description of the approach and its application to the empirical data they use is great, but I could not envision how to actually use this method for my own work. I know that data and code would presumably be available upon acceptance for publication, but the authors should consider how they plan (if at all) to make this method readily implementable by fellow researchers. Doing so would greatly strengthen the impact of this paper and bring about the transformative change the authors describe. As it presently stands, this approach brings together various ways of thinking into a novel conceptual approach, but not a novel practical approach. I sincerely hope it becomes the latter as well.

Our response: This is a great point. We have packed our method into an R script and uploaded at a public site <https://github.com/CCBBeijing/PPMultilayerNetwork>, from which researchers worldwide can freely download and use our method. We will make every effort (including presentations to the conferences, seminars, and workshops) to disseminate our method and

publicize its use to a range of biological, evolutionary, and medical fields.

My minor comments are below:

We are very much grateful for this reviewer to read and check our manuscript so carefully. The following editorial comments have all been addressed.

4) At various places in the manuscript the font color changes unexpectedly from black to a more grayish color. Presumably, this is the result of editing or copying from other drafts/communications.

Our response: This has been cleaned up, making the manuscript consistent in font and color.

5) L129-130 Please related this description of developmental plasticity more explicitly to the preceding sentence. Related to part of my point 1 above, plasticity was simply the difference between control and stress and then this was taken at various timepoints in development, correct?

Our response: In this revised version, we define developmental phenotypic plasticity before this concept is focused. It is correct that developmental plasticity is defined as the difference of abundance at different time points between control and stress. We have made this definition clearer in this version.

6) L131 Please define or describe what composite functional mapping is and if it is unique to your framework.

Our response: We have explained this approach in the Methods section.

7) L172 Change “A total of” to “The”

Our response: We have changed it.

8) L173 and ‘than’ after the word ‘rather;

Our response: We have changed it.

9) L176 M12 is listed twice

Our response: We have corrected it.

10) L189 How were these 45 links identified as actually occurring

Our response: We have modified this sentence.

11) L191 Is it SNPs or modules at this point?

Our response: We have corrected it as modules.

12) L196 change ‘that trigger’ to ‘have’

Our response: We have changed it.

13) L197-198 This is a cool level of detail to be able to get to!

Our response: Thank you for you encouraging comment.

14) L223-224 This predictive and hypothesis-driving potential could be very useful

Our response: Thank you for you encouraging comment.

15) L242 ‘receive more incoming links that send outgoing links’

Our response: We have changed this sentence as suggested.

16) L256 remove ‘we actually have’

Our response: We have removed it.

17) L266 remove ‘how’

Our response: We have removed it.

18) L276 ‘leadership’

Our response: We have corrected it.

19) L411 italicize *S. aureus*

Our response: We have done it.

20) L416 italicize *S. aureus*

Our response: We have done it.

21) L449 remove 'but'

Our response: We have corrected it.

22) L459-460 Remove 'published...University' and just include the citations

Our response: We have modified it.

23) Methods I know the journal dictates manuscript formatting, but I think this manuscript could greatly benefit from having the methods before the results or co-mingled. This is especially important because of the tool-like nature of this manuscript. Making it easier to see how what was done relates to what was found would be very useful for this manuscript.

Our response: We understand this concern. We have provided the flowchart of method derivation at a beginning of the Results section so that the readers can better understand the idea of our method derivation.

24) Figure 1A rightmost panel: I am not sure what to make of this figure. Is it meant to show how plasticity was calculated? If so, I am still confused and perhaps more description is warranted.

Our response: This figure shows how the same strain grows differently between control and stress. If the growth trajectory of a strain is on the diagonal, then this indicates that this strain has the same growth between the two environments, i.e., no phenotypic plasticity.

Reviewer #2 (Remarks to the Author: Reproducibility):

As indicated elsewhere in my review, I think the methods could be made more user-friendly and hands-on so that those not deeply involved in the concepts/theory could implement the approach.

Our response: We have packed our method into an R script and uploaded at a public site <https://github.com/CCBBeijing/PPMultilayerNetwork>. from which researchers worldwide can freely download and use our method. We will make every effort (including presentations to the conferences, seminars, and workshops) to disseminate our method and publicize its use to a range of biological, evolutionary, and medical fields.

Reviewer #3 (Remarks to the Author: Overall significance):

Review of Inferring multilayer interactome networks shaping phenotypic plasticity and evolution
by Yang, Jin, He, and Wu

Main comments

Yang et al. attempt to deduce networks of interactions between loci underpinning a plastic trait. Their goal is laudable and ambitious, and the experimental aspect of their work is fascinating on its own. However, this manuscript, in its current form, suffers from a number of shortcomings and, in my view, requires extensive revisions before it reaches a publishable unit. Below, I highlight some of the major shortcomings of the manuscript in its current form.

Our response: Thank you for your constructive comments and critiques.

Reviewer #3 (Remarks to the Author: Strength of the claims):

First, the premise of the experiment is not well-justified. Specifically, how does one disentangle the plastic response from the genetic response, given the fact that GWAS, by definition, is linking phenotypic variance directly to the genetic variation? In the absence of such justifications, statements such as "... tremendous inter-strain variation in the pattern ... implies the existence of genes that govern phenotypic plasticity." (lines 123-134) are not warranted.

Our response: In this revised version, we used a flowchart to show the structure of genetic data and phenotypic data. The phenotypic data are longitudinal, measured in two environments. The same strain may have the same or different growth between the two environments, with the degree of difference defined as phenotypic plasticity. Fig. 2A shows that different strains may have different levels of phenotypic plasticity. Thus, by viewing phenotypic plasticity as a (composite) complex trait, we map the genetic architecture of phenotypic plasticity. All these statements have been made clear in the text.

Second, the computational model to infer the so-called omnigenic interactome networks, which is the crux of the manuscript, is explained in vague terms. In the introduction section (lines 81 - 94), evolutionary game theory and network theory are invoked, without even the most cursory explanation as to why these theories were utilized to infer interactions from QTLs deduced from a GWAS study. The method section does very little to illustrate how the model works. Statements such as "according to network theory" (line 775) are extremely confusing as they imply that the network theory is something akin to the second law of thermodynamic, while in fact, the sentence is as much useful as a sentence that starts with "according to the theory of evolution ...". More precise language and more extensive explanations of the model are required.

Our response: This is a reasonable comment. In this version, we have clearly described our model and its derivation procedure in the Methods section. We have added a section – Rationale

of Network Reconstruction to explain how evolutionary game theory can be used to dissect complex traits.

Third, the section title “Detection of network communities” (lines 153 - 177), while being crucial for understanding the manuscript, lacks the necessary explanations and elaboration. The developmental modularity theory is introduced without any explanations. (as side note: I presume that BIC (line 166) stands for Bayesian information criterion, but acronyms should always be introduced properly.)

Our response: We have elaborated the principle of developmental modularity theory to break down a large network into its distinct network communities. Please see the **Detection of network communities** section.

Finally, it is not at all clear how measuring genetic effects over 50 hours is a justifiable method to deduce the gene regulatory network that underlies a phenotype when the population in question has undergone roughly 100 generations already (assuming generation time of 30 min). How does adaptive evolution in the presence of antibiotics affect the topology of the interactome networks inferred from the QTLs? In the absence of such justifications, the biological nature and significance of the inferred interactions, such as the supposed “antagonistic” relationship between QTL2728755 and NP2748895, remains elusive. The extensive discussion about the nature of interactions between QTLs and SNPs, as promoters or suppressors, is similarly affected.

Our response: This is a good question. It is possible that bacteria have undergone many generations. However, our question is to map how genes bacteria currently carry determine their growth trajectories, no matter how they generations have passed. It is plausible that the model is methodologically extended to cover the effect of genetic mutations on growth trajectories. We have discussed this issue. For many other organisms without rapid mutations, the current model can be directly used with no problem.

Aside from the aforementioned comments, given the complexity of the model proposed in this manuscript, it's a pity that its poor structure prevents an adequate comprehension and a fair assessment of its central claim.

Our response: We have reorganized the structure of the Methods. We believe that, along with a flowchart (Fig. 1), the new structure has made our model derivation a much clearer and more smooth procedure.

Minor points

Line 70: Suggest citing some of the famous papers on the definition of epistasis, including

Patrick Philips' The language of gene interaction (GENETICS July 1, 1998 vol. 149 no. 3 1167-1171).

Our response: This reference is cited.

Line 71: “statistical approaches used in the literature cannot reveal the casualty of epistasis and the sign casualty” => causality not casualty.

Our response: This has been corrected. Thanks!

“..., consistently with a larger amount of growth ...” (Line 116) is a slightly misleading characterization of the growth curves in Figure 1, as most strains reach similar levels of abundance under stress compared to the control.

Our response: This has been clarified.

Ref 43 should be Maynard Smith, J. and not Smith, J. M.

Our response: This has been corrected.

In “ ... implying their neglected role ...” (144) => probably “negligible” and not “neglected”.

Our response: This has been corrected. Thanks!

Line 455: It is either “a wealth of literature” or “a body of literature”, but not both.

Our response: This has been corrected. Thanks!

Line 682: “differences in equation (1)” => “difference equation (1)”

Our response: This has been modified.

Line 752: “a hybrid of the EM algorithm” => a hybrid of the expectation–maximization (EM) algorithm”

Our response: This has been changed.

Reviewer #4 (Remarks to the Author: Overall significance):

In this paper, the authors analyse the genetic basis of *S. aureus* resistance to vancomycin by using a statistical approach based on the omnigenic model. They measured growth curves of 99 strains with and without this stress, and used a genome-wide association method for time series data to find significant core SNPs controlling this response. The authors then applied evolutionary game theory to construct an interacting network of core SNPs which can explain the measured response curves. Whereas a ‘typical’ genetic model of this phenotype would consist solely of the five independently significant QTLs discovered in the first phase; in this model, insignificant SNPs typically have large individual effects on the phenotype, but most of these cancel out due to regulatory effects from other core SNPs in their network. These interactions can be measured and, in some cases, interpreted. The authors imply that many of the regulatory SNPs would be common actors in a range of phenotypes, though this is not tested.

The omnigenic theory of trait inheritance is an interesting one, worthy of further study in species other than humans. I believe that the statistical approaches and experiments presented here are broadly appropriate to address this question in a limited way. However, with current manuscript I had the following issues with its presentation:

- Broad, sweeping claims are made about the approach being a ‘paradigm-shifting’ framework. However, only a single phenotype, in a single species, in a specific strain collection was tested. A key feature of the omnigenic model is that the regulatory SNPs regulate many phenotypes, but as the authors only test one phenotype they cannot identify whether the links in their regulatory networks are shared.

Our response: Thank you for your time and effort to evaluate our work. Existing approaches for mapping or association studies are based on reductionist thinking; i.e., they characterize individual key QTLs by testing the significance of a SNP at a time. Our model can analyze all SNPs at the same time, by which we can chart a global picture of the genetic architecture that involve all loci in a GWAS, allowing their possible interactions in a network. As such, compared to traditional approaches, our model can be considered as a holistic one. If our model is published, it could make a paradigm shift for GWAS from reductionist thinking to holistic thinking.

However, we do not deny traditional approaches, rather capitalize on and expand the results from traditional approaches to extract information that is unobserved by traditional approaches.

This reviewer is right. A computational model can better reveal regulatory networks if multiple traits are included. However, a model for reconstructing multi-trait omnigenic networks would be extremely difficult or even impossible without a clear understanding of model derivation for a single trait. Our work here provides a general framework for SNP network reconstruction, which can be extended, with aid of knowledge from more disciplines, to more complex but more biological meaningful situations.

- The omnigenic model is essentially taken as a prior for the analysis here. All SNPs are included from the bottom-up, as a feature. But what about other models? Does this model the data better than say a model with the QTLs? Or a polygenic model such as GCTA and LDAK? Without these comparisons, I do not think that the authors can claim that their model and approach is superior in terms of predictive power or mechanistic understanding of these traits.

Our response: Traditional polygenic model assumes that a trait is controlled by many genes each with a minor additive effect, although the number of genes involved is generally unknown. However, our model does not rely on this assumption; i.e., it allows genes to act in any manner. A gene may have a large or small effect, it may or may not interact with other genes, and the model coalesces all genes into a unified network.

The most relevant model we can compare our network model to is QTL mapping model. In our manuscript, we have compared our model with traditional QTL model. Below is a diagram which shows the positions of traditional QTL model with our network model. As can be seen, our network model is the prolongation of traditional QTL model (FunMap and coFunMap). In other words, our model does not refuse traditional QTL model, but uses and integrate it with other disciplines to extract and excavate more information hidden in the genetic architecture of complex traits.

- There is not enough clarity, particularly in the introduction and conclusion, on what it actually

is that the authors are testing. There are a lot of buzzwords, but from my reading the main thing this manuscript does show us one way to construct an omnigenic model for a trait, and compare it with some of the QTLs found in a GWAS. I would encourage the authors to spend more time introducing monogenic, polygenic, omnigenic models, and missing heritability. Give the biological/genetic interpretations of each, and compare and contrast them. Additionally, note the difference between phenotype prediction and understanding the biological basis of phenotypes – two related but different goals of these models.

Our response: The motivation to derive our model is to systematically map QTLs by including all SNP in a GWAS. Phenotypic prediction is out of scope of our work.

In the Introduction, we have provided an overview of the current status of QTL mapping and association studies, followed by the motivation of our model development. We do not think that the monogenic model is immediately relevant to our model because we are interested in quantitative traits.

- There is a long part of the paper and the figures showing and describing links in networks. Without either strong evidence that this model is superior, or biological interpretation (SNPs in this section are simply numbered), this is both speculative and offers poor explanatory power.

Our response: This is a good comment. In this revised version, we have performed extensive gene enrichment analysis and pathway analysis. In many places, we have demonstrated the biological relevance of genetic interactions detected by our model. For example, our model shows that QTLs Q550323 and Q183488 have large independent genetic effects. It is found that these QTLs are located at candidate genes (*SdrC* for Q550323 and *oppA* for Q183488) that are directly involved in antibiotic resistance, which thereby explains the reason why they have remarkable independent effects. Also, several SNP-SNP interactions detected by our model can be well annotated to protein-protein interactions through KEGG analysis. In other words, a pair of proteins encoded by linked genes in SNP networks reconstructed by our model are correspondingly linked in protein networks. These lines of evidence can well explain the validity of our model.

- The section on ‘model validation’ was not particularly convincing. It mostly seemed to repeat the same experiment, as the essentially the same phenotype and analysis were redone. This felt like a bit of a missed opportunity: if the authors had made mutants which can directly measure effects of SNPs, they could have tested regulatory predictions from the constructed networks, and compared these with a model of individual QTLs. At least, comparing predictive power of the network approach versus the QTL approach in a new dataset would have been more valuable.

Our response: We have adopted this comment by removing this part from the manuscript. In this revised version, we have added an independent GWAS experiment using the same species,

testing the genetic architecture of phenotypic plasticity to species coexistence. The two different experiments provide highly related but also highly complementary results to validate our model.

Reviewer #4 (Remarks to the Author: Impact):

I do think that this paper could be an interesting addition to the microbial statistical genetics literature. As far as I am aware, no one has previously explored the omnigenic model in bacteria before. But to do so effectively, the authors need to be less bold in their claims, clearly compare the genetic models they are testing, and be refocus to target a comparison between omnigenic and polygenic models specifically.

Our response: Thank you for your constructive comments. We have compared our model with the most relevant QTL mapping model.

Overall, I can see the rationale for this model, but is based on asserting that all SNPs have an effect: can we connect these in a way to get the pattern in this (single) GWAS? I don't see evidence here that this model refutes just the significant QTLs, or a standard polygenic model, predicting the data equally well.

Our response: This reviewer is right. Our network model is the prolongation of traditional QTL model. In other words, traditional QTL model stops at a point which is far from a destination, but our model moves further to approach the destination.

We have used and integrated results from traditional approaches with other disciplines to extract and excavate more information hidden in the genetic architecture of complex traits.

For a more impactful paper which would inspire others to use this model, I would expect to see more targeted experiments, using new strains or mutants, which tested these models predictive and explanatory power. I would also expect to see more reusable code, which include some continuous integration testing.

Our response: This is a good point. The motivation of this work is to develop a new model, with real-world experiments that provide a proof of concept for the model. We are closely collaborating with microbiologists to test and edit key genes and regulations detected by our model. We have packed our method into an R script and uploaded at a public site <https://github.com/CCBBeijing/PPMultilayerNetwork>, from which researchers worldwide can freely download and use our method.

Reviewer #4 (Remarks to the Author: Strength of the claims):

Required:

1) The effects of population structure are dealt with only in passing, but are a major issue for many GWAS studies. I agree that the QQ plot and Manhattan plot look broadly correct, but would need to also see:

a. A phylogenetic tree of the isolates, ideally with some summary of the phenotype mapped on as a trait.

Our response: We have built up the phylogenetic tree of the strains used in our GWAS (Fig. S8).

b. More information of how the eight subpopulations were identified and removed.

Our response: We have provided more information about population structure and phenotype adjustment (In the Methods section).

c. A comparison of a standard linear mixed model versus a quantitative summary statistic of the trait (for example the growth rate and/or carrying capacity)

Our response: A linear mixed model for GWAS has been developed static traits, but has not been developed for longitudinal traits. Therefore, a comparison with a linear mixed model is not available for this study.

2) More biological inference of the *S. aureus* results integrated into the main text, moving table S1 into the main results, and describing SNPs in terms of known function rather than just position. In particular, a biological interpretation of the modules of SNPs would make the model more convincing.

Our response: We have provided extensive gene enrichment analysis of modules, QTLs, and SNPs and extensive pathways analysis of networks. We understand that these are important pieces of information for our manuscript. Yet, because the manuscript is already too lengthy, we only can put this gene annotation table as a supplementary table.

3) Make a clear comparison between the omnigenic model of trait heritability, and a mono/polygenic model. Show which has better explanatory power. I would note that more sub-significant SNPs affecting a phenotype is well established and modelled by e.g. the GCTA model. Implicitly, the GCTA model assumes that all SNPs have small effects. So the difference with the omnigenic model is that many have large effects, but they cancel out through interactions. But, which is the model better supported by the data here?

Our response: The motivation to derive our model is to systematically map QTLs by including all SNP in a GWAS. The most relevant model we can compare our network model to is QTL

mapping model. The GCTA model aims to address the missing heritability problem. For this reason, we have made the comparison of our model with QTL mapping model.

4) In missing heritability, calculating h^2 , and h^2 from the QTLs would be a really helpful way to compare these models. Adding the significant SNPs from the module approach to a h^2 calculation would also be informative.

Our response: This is an excellent question. Currently, we can only postulate the contribution of regulatory networks to missing heritability, but do not know how to quantify this contribution. This is certainly our next project to pursue.

5) Likewise, using a method such as SpydrPick or just simple associations of interaction terms to find epistasis via ‘conventional’ means would be useful to compare to interactions in the networks (even if the result is that this is underpowered).

Our response: These pairwise epistatic analyses are based on the marginal distribution of SNP genotypes. As explained above, our model based on a joint analysis of all SNPs is the prolongation of these traditional approaches, producing results that cannot be detected by the latter. For example, we can detect dependent effects of a SNP, whereas traditional approaches can only detect its net (overall) effect.

6) The cancelling out of strong effects through interactions in the omnigenic model is necessary to reconcile individual effects at the lower level with no effect overall (the QTLs in the GWAS). What would the alternative look like? Is there a way to not have an omnigenic model and still get the GWAS effects?

Our response: Existing GWAS approaches are based on marginal genetic analysis; i.e., detecting a significant SNP unconditional upon all other SNPs. These approaches cannot detect a SNP whose overall effect is not significant but independent effect is remarkable.

Optional

1) The authors only examine SNPs in the core genome, but the accessory genome is known to have an important role in adaptive response in bacteria (e.g. <http://dx.doi.org/10.1038/s41559-017-0337-x>, and many others). Could the accessory genome variation be included in an omnigenic model?

Our response: This has no problem. The model can handle any type and any number of genetic variants.

2) If the same regulatory loci and interactions were found to hold for other phenotypes in the species, the model and approach here would be more convincing. A multi-phenotype study,

perhaps RNA-seq, would be most powerful of all. There are some public datasets available with this kind of data.

Our response: To test our model, we design and conduct two real-world experiments. As a proof of concept, we analyze a trait - growth. Multiple traits can be incorporated, but as mentioned above, which needs more extensive statistical derivations, beyond the scope of this manuscript.

3) Validation of the model by creating mutants based on the networks speculated above, to test whether these SNPs do affect regulation would add a lot. For example, can both the network approach, and QTL approach be used to predict phenotype response in a recombinant mutant. Where these predictions are divergent, then testing which phenotype the mutant has would be a powerful confirmation of that model.

Our response: The motivation of this work is to develop a new model, with real-world experiments that provide a proof of concept for the model. We are closely collaborating with microbiologists to test and edit key genes and regulations detected by our model.

Additional minor comments:

1) In the introduction: omnigenic theory is describe as all traits being controlled by all genes. My understanding is that there are common regulatory genes which are involved in many phenotypes, not that every gene is functional. How could the authors reconcile their assertion that all genes are involved with the frequent loss of genes in bacteria?

Our response: Our model can analyze all loci (no matter what they are) at the same time. Model derivation is based on a specific trait. Multi-trait analysis is doing, but needs more extensive work.

2) Network is used many times with a specific definition in the introduction.

Our response: Network is a common concept. We provide an explanation of it in the Introduction.

3) ‘millions of millions of gene pairs’ – in a typical bacterial population there would be around 10^7 genes. Be more specific if possible. Also, this is consistently stated to be computationally intractable. Networks with this many edges are possible to analysed however.

Our response: To handle this high-dimensional issue, we introduce developmental modularity theory. This can help us handle any number of loci.

4) Inferring this many interactions from a few samples is still likely to be underpowered, even with improved analysis (even though the time series does add power). How confident are the authors that their network is the only correct one?

Our response: This issue has been discussed in our previous studies. The characterization of dependent effects is not dependent on sample size but on net genetic effect curves. The estimation of main genetic effect curves does rely on sample size, but both simulation and experimental studies suggest that a sample size of 100 to 400 can reasonably well estimate such curves if the number of time points is two or more times the number of curve parameters (Zhao et al. 2005; Sang et al. 2019).

Zhao, W., Hou, W., Littell, R. C. & Wu, R. Structured antedependence models for functional mapping of multivariate longitudinal traits. *Stat. Methods Mol. Genet. Biol.* 4(1), 33 (2005).

Sang, M. et al. A dissection model for mapping composite traits. *Plant J.* 97, 1168-1182 (2019).

5) ‘Statistical approaches cannot infer the causality of epistasis’. Is this specific to epistasis? Approaches such as Mendelian randomization, or that of Pearl et al have attempted this I believe. Also a typo in ‘causality’.

Our response: Traditional approaches can only estimate part of the features of epistasis, such as strength, sign, and direction. Our model can fully characterize bidirectional, signed and weight genetic interactions. The typo has been corrected.

6) ‘design an efficient and effective gene editing program’. In bacteria? More broadly?

Our response: Our model is generic, but it is validated by bacterial data. Therefore, results from our model are not limited to bacteria but to any other species (including humans).

7) A deficiency of linear additive models is stated as missing random effects, but linear mixed models, the most commonly used models in GWAS and breeding, do include random effect terms.

Our response: Yes, random effects can be included, but they cannot be used to characterize the strength and causality of genetic interactions among individual pairs.

8) Is bDSW a standard acronym? I found its use confusing, and would suggest that simply ‘edges’ or ‘interactions’ would be easier to follow.

Our response: Yes, this is a standard acronym. ‘Edges’ or ‘interactions’ are less informative than bDSW. More importantly, its use does not affect the understanding of our model.

9) I noted that the genetic effect vs time curves mostly show effect during the growth phase (rather than stationary). Is this because this is where phenotype difference/derivative with respect to time is greatest, and there is greatest power?

Our response: The genetic effects are estimated from growth curves at the two alternative genotypes. Therefore, they can be in any form, depending on genotypic curves.

10) ‘find that their effect curves are quite flat, implying their neglected role in shaping phenotypic plasticity.’ How does this implication follow?

Our response: This has been changed as “find that their effect curves are close to zero,”

11) I appreciate that minimizing BIC is a good quantitative way to choose the number of modules, but it might be informative to use a lower number, and see if any obvious biological pattern is apparent (as is, many of the module effects look very similar).

Our response: Although they look similar, their differences are large enough to classify them into different modules.

12) The beginning of the section ‘multilayer genetic networks’ could more clearly and succinctly be described as ‘we recursively subdivide communities’.

Our response: This sentence contains too much information that the suggestion ‘we recursively subdivide communities’ cannot capture it.

13) On sparsity: is finding a sparse network a necessary property the method of subdivision, or a finding when applied to this dataset?

Our response: Sparsity is basic rule of biological networks.

14) ‘Most links are directional rather than reciprocal’ – invoking interactions between populations of animals is unlikely to be universal, does it really apply to SNPs and antimicrobial resistance?

Our response: SNP interactions form a micro-community that obeys a general rule of interactions.

15) The discussion restates a lot of the results, and could be trimmed. Some of the biological interpretation should be in the results.

Our response: In the Results, we focus on the description of our findings, whereas in the Discussion, we discuss the relevance and implications of these findings. Also, the Results section is already very long, thus to balance, some information is better given in the Discussion.

16) ‘Our model integrates the theories of two recent perspective articles published in Cell by J. Pritchard and M. Snyder both at Stanford University’. The citations will suffice.

Our response: This has been changed.

Reviewer #4 (Remarks to the Author: Reproducibility):

1) The code URL isn’t listed in the manuscript, but I found it (with a typo) in the reporting summary. In such a manuscript, the code needs to be more prominent, and in a state where others can use it. At the moment it is fairly unstructured, has local links to files, and could certainly not be reused or replicated by others.

Our response: The code URL has now been given in the current version.

2) Analysis in this paper was written from scratch, no existing software was used. How can we be sure that no errors occurred in this process? The final model is complex. Some testing of the code itself is needed.

Our response: This is a technical issue. We ensure that our analysis is statistically and computationally correct. As a computational biologist, we perform many simulations to validate the correctness of the model.

Reviewer comments, second version:

Reviewer comments:

Reviewer #1 (Remarks to the Author: Overall significance):

The approach authors have taken to map the genetic effects, and relationships between them, on a bacterial phenotype is novel and interesting. The authors do a good job at citing previous methodological papers, and have now cited relevant literature on the genetic basis of vancomycin non-susceptibility in *Staphylococcus aureus*.

Reviewer #1 (Remarks to the Author: Impact):

This paper may influence the way genetic mapping of bacterial phenotypes is approached in the future.

Reviewer #1 (Remarks to the Author: Strength of the claims):

All my major and minor comments have been successfully addressed by the authors.

Reviewer #1 (Remarks to the Author: Reproducibility):

The authors have made an important effort to make their code and raw data, including bacterial growth measurements and sequencing data, available. They also provided information on bioinformatics tools that were previously missing. They have successfully addressed my comments related to data availability.

Reviewer #2 (Remarks to the Author: Overall significance):

This manuscript proposes a novel method of disentangling gene regulatory networks from time series genetic mapping (QTL/GWAS) data. It is applied and discussed with respect to understanding the developmental and genetic bases of phenotypic plasticity and is presented via analysis of bacterial resistance to vancomycin and competition. The method proposed seems to better capture gene interactions than current approaches to analyzing mapping data.

I am excited about this method and its potential applicability to a diversity of traits and environments. I think it could constitute a major methodological advance for linking genotypic and phenotypic variation.

Reviewer #2 (Remarks to the Author: Impact):

I think this paper will certainly influence thinking in the field and may pave the way for additional sophistication in how we associate phenotype and genotype. If nothing else, it will provoke others to think critically about existing methods and how they compare to this approach---such reflection and discourse is a benefit for the community.

Reviewer #2 (Remarks to the Author: Strength of the claims):

I am happy with the authors' efforts to address my concerns and have no new issues to raise.

Reviewer #2 (Remarks to the Author: Reproducibility):

I am happy with the authors' decision to make their method available through R and that they have

included relevant links/data that were missing the first draft.

Reviewer #4 (Remarks to the Author: Overall significance):

In this revision, the authors add clarifications to the text, analyse a further phenotype, add links to their code and data. This has improved the clarity of the paper, and the overall significance remains the same. I found the addition of figure 1 particularly helpful.

Reviewer #4 (Remarks to the Author: Impact):

Based on the revision, and the previous editorial comments, Communications Biology would be the best fit. I do appreciate that another phenotype has been analysed, but this did not validate the model against existing approaches.

Reviewer #4 (Remarks to the Author: Strength of the claims):

This revision is an improvement. One key point I would encourage the authors to add, is that the advance of this paper is to take results from CoFunMap/QTL mapping, and then use a range of approaches to better interpret them in light of the omnigenic model.

I will focus on two of my previous comments which were marked 'essential' by the editors:

- Comment #2: benchmark omnigenic vs polygenic models. I agree with the authors that comparing with the standard (reductive) QTL model is probably most appropriate. The issue is then how do we then know this (holistic) network module model represents true biology better than using the individually significant QTLs. Reading each section, both seem to provide plausible biological findings. But, it is easy to come up with convincing post hoc justifications for any association. Which is best, and how can we know? I cannot think of a way to do this, other than by experimentation.

Please could the authors comment on this in the text.

Relatedly, I would still find a comparison with an LMM approach using either fitted growth rate/carrying capacity (thus avoiding a longitudinal trait) useful here.

- Comment #4/12, missing heritability. As a rough estimate, why not try calculating R^2 in a linear model vs. growth rate, using selected SNPs (from QTL mapping) or modules. It may also be possible to calculate variance explained in a lasso regression against longitudinal data.

Reviewer #4 (Remarks to the Author: Reproducibility):

Code is available, and seems to run. Though I would encourage not zipping it, so it is searchable and readable on github.

Author rebuttal, second version:

Reviewer #1 (Remarks to the Author: Overall significance):

The approach authors have taken to map the genetic effects, and relationships between them, on a bacterial phenotype is novel and interesting. The authors do a good job at citing previous methodological papers, and have now cited relevant literature on the genetic basis of vancomycin non-susceptibility in *Staphylococcus aureus*.

Reviewer #1 (Remarks to the Author: Impact):

This paper may influence the way genetic mapping of bacterial phenotypes is approached in the future.

Reviewer #1 (Remarks to the Author: Strength of the claims):

All my major and minor comments have been successfully addressed by the authors.

Reviewer #1 (Remarks to the Author: Reproducibility):

The authors have made an important effort to make their code and raw data, including bacterial growth measurements and sequencing data, available. They also provided information on bioinformatics tools that were previously missing. They have successfully addressed my comments related to data availability.

Our response: Thank you for your encouraging comments.

Reviewer #2 (Remarks to the Author: Overall significance):

This manuscript proposes a novel method of disentangling gene regulatory networks from time series genetic mapping (QTL/GWAS) data. It is applied and discussed with respect to understanding the developmental and genetic bases of phenotypic plasticity and is presented via

analysis of bacterial resistance to vancomycin and competition. The method proposed seems to better capture gene interactions than current approaches to analyzing mapping data.

I am excited about this method and its potential applicability to a diversity of traits and environments. I think it could constitute a major methodological advance for linking genotypic and phenotypic variation.

Reviewer #2 (Remarks to the Author: Impact):

I think this paper will certainly influence thinking in the field and may pave the way for additional sophistication in how we associate phenotype and genotype. If nothing else, it will provoke others to think critically about existing methods and how they compare to this approach--such reflection and discourse is a benefit for the community.

Reviewer #2 (Remarks to the Author: Strength of the claims):

I am happy with the authors' efforts to address my concerns and have no new issues to raise.

Reviewer #2 (Remarks to the Author: Reproducibility):

I am happy with the authors' decision to make their method available through R and that they have included relevant links/data that were missing the first draft.

Our response: Thank you for your encouraging comments.

Reviewer #4 (Remarks to the Author: Overall significance):

In this revision, the authors add clarifications to the text, analyse a further phenotype, add links to their code and data. This has improved the clarity of the paper, and the overall significance remains the same. I found the addition of figure 1 particularly helpful.

Our response: Thank you for your encouraging comments.

Reviewer #4 (Remarks to the Author: Impact):

Based on the revision, and the previous editorial comments, Communications Biology would be

the best fit. I do appreciate that another phenotype has been analysed, but this did not validate the model against existing approaches.

Our response: This is a Methods paper. As you know, a Methods paper requires a different level of biological validation than a data paper. Traditional Methods papers justify their models by reanalyzing and reinterpreting published datasets. Our justification has gone above and beyond the traditional standard by designing and conducting two new experiments. We believe that two different datasets from the well-controlled experiments of the same species can very well justify the method we proposed. Given its interdisciplinary nature, this paper is more appropriate for Nature Communications than Communications Biology.

Reviewer #4 (Remarks to the Author: Strength of the claims):

This revision is an improvement. One key point I would encourage the authors to add, is that the advance of this paper is to take results from CoFunMap/QTL mapping, and then use a range of approaches to better interpret them in light of the omnigenic model.

I will focus on two of my previous comments which were marked 'essential' by the editors:
- Comment #2: benchmark omnigenic vs polygenic models. I agree with the authors that comparing with the standard (reductive) QTL model is probably most appropriate. The issue is then how do we then know this (holistic) network module model represents true biology better than using the individually significant QTLs. Reading each section, both seem to provide plausible biological findings. But, it is easy to come up with convincing post hoc justifications for any association. Which is best, and how can we know? I cannot think of a way to do this, other than by experimentation.

Please could the authors comment on this in the text.

Our response: We agree with this statement. The best way to validate a model is to conduct experiments as we did. This is not only appropriate for our mode, but for all other models, including LMM.

Relatedly, I would still find a comparison with an LMM approach using either fitted growth rate/carrying capacity (thus avoiding a longitudinal trait) useful here.

Our response: Thank you for your agreeing with us that our model can be compared with QTL models more appropriately than with polygenic models. However, we also highly respect the reviewer's suggestion to compare the model with the LMM approach.

As you know, an LMM is developed for static traits, lacking a strong capacity to handle dynamic traits. Our model is derived on dynamic traits, making a complete comparison a difficult endeavor. The reviewer thus suggests using fitting a growth parameter (we choose growth rate). We implement one of the LMM models - TASSEL to analyze growth rate data and estimate the heritabilities of individual SNPs.

Peter J. Bradbury, Zhiwu Zhang, Dallas E. Kroon, Terry M. Casstevens, Yogesh Ramdoss, Edward S. Buckler, TASSEL: software for association mapping of complex traits in diverse samples, *Bioinformatics*, Volume 23, Issue 19, 1 October 2007, Pages 2633–2635.

However, this comparison should be cautious. First, LMM uses only one feature of growth curves, whereas our model capitalizes on full growth curves. This comparison is not very fair. Second, LMM can only identify a single significant SNP at a time, whereas our model analyzes all SNPs simultaneously from which all possible pairwise interactions can be detected once at a time. Therefore, our model and LMM interpret the data from different perspectives.

- Comment #4/12, missing heritability. As a rough estimate, why not try calculating R^2 in a linear model vs. growth rate, using selected SNPs (from QTL mapping) or modules. It may also be possible to calculate variance explained in a lasso regression against longitudinal data.

Our response: This is a very good point. It can test whether a missing heritability issue exists for general GWAS.

Because our experiment has replicates (e.g., control vs. stress, monoculture vs. con-culture), we can estimate the heritability of growth rate. In the abiotic phenotypic plasticity experiment, the overall heritability of growth rate is 0.763, but the total heritability explained by all QTLs is 0.516 in control and 0.021 in stress, leaving 0.247 and 0.742 unexplained. In the biotic phenotypic plasticity experiment, the overall heritability of growth rate is 0.602, but the total

heritability explained by all QTLs is 0.170 in monoculture and 0.335 in co-culture, leaving 0.432 and 0.267 unexplained.

Both experiments suggest that there is a plenty of missing heritability, indicating the possibility and necessity to excavate those hidden genetic variants. Our model provides a unique way to retrieve missing heritability.

We have incorporated the above results into the manuscript (highlighted in red).

Reviewer #4 (Remarks to the Author: Reproducibility):

Code is available, and seems to run. Though I would encourage not zipping it, so it is searchable and readable on github.

Our response: We have uploaded all code used at each step of data analysis at <https://github.com/CCBBeijing/PPMultilayerNetwork>.

Reviewer comments, third version:

REVIEWERS' COMMENTS:

Reviewer #4 (Remarks to the Author: Overall significance):

The authors have added a section on LMMs and heritability in response to my second review. I have no further comments or suggestions.

Reviewer #4 (Remarks to the Author: Reproducibility):

I note that the code is still in three separate .zip files. It would make it easier for others to use if these were unzipped and the R scripts these archives contain uploaded directly. You don't strictly need to do this, but it does help with code re-use.

Author rebuttal, third version:

We have addressed the editorial requests in the author checklist (in a separate pdf file). Below we also provide a response to the reviewer's comment.

Reviewer #4 (Remarks to the Author: Overall significance):

The authors have added a section on LMMs and heritability in response to my second review. I have no further comments or suggestions.

Reviewer #4 (Remarks to the Author: Reproducibility):

I note that the code is still in three separate .zip files. It would make it easier for others to use if these were unzipped and the R scripts these archives contain uploaded directly. You don't strictly need to do this, but it does help with code re-use.

Our response: We have unzipped the files. We have provided a DOI for the data and code as [10.5281/zenodo.4951175](https://doi.org/10.5281/zenodo.4951175) in the GitHub repository.

All the changes to the text are highlighted in red.